# The World of Cyclic Dinucleotides in Bacterial Behavior

**DOI:** 10.3390/molecules25102462

**Published:** 2020-05-25

**Authors:** Purificação Aline Dias da, Azevedo Nathalia Marins de, Araujo Gabriel Guarany de, Souza Robson Francisco de, Guzzo Cristiane Rodrigues

**Affiliations:** Department of Microbiology, Institute of Biomedical Sciences, University of São Paulo, São Paulo 01000-000, Brazil; aline.purificacao@usp.br (P.A.D.d.); nathalia.marins.azevedo@usp.br (A.N.M.d.); g.araujo@usp.br (A.G.G.d.); robfsouza@gmail.com (S.R.F.d.)

**Keywords:** GGDEF, SMODS, DAC, c-di-GMP, cGAMP, c-di-AMP

## Abstract

The regulation of multiple bacterial phenotypes was found to depend on different cyclic dinucleotides (CDNs) that constitute intracellular signaling second messenger systems. Most notably, c-di-GMP, along with proteins related to its synthesis, sensing, and degradation, was identified as playing a central role in the switching from biofilm to planktonic modes of growth. Recently, this research topic has been under expansion, with the discoveries of new CDNs, novel classes of CDN receptors, and the numerous functions regulated by these molecules. In this review, we comprehensively describe the three main bacterial enzymes involved in the synthesis of c-di-GMP, c-di-AMP, and cGAMP focusing on description of their three-dimensional structures and their structural similarities with other protein families, as well as the essential residues for catalysis. The diversity of CDN receptors is described in detail along with the residues important for the interaction with the ligand. Interestingly, genomic data strongly suggest that there is a tendency for bacterial cells to use both c-di-AMP and c-di-GMP signaling networks simultaneously, raising the question of whether there is crosstalk between different signaling systems. In summary, the large amount of sequence and structural data available allows a broad view of the complexity and the importance of these CDNs in the regulation of different bacterial behaviors. Nevertheless, how cells coordinate the different CDN signaling networks to ensure adaptation to changing environmental conditions is still open for much further exploration.

## 1. Introduction

In the mid-2000s, the idea emerged that c-di-GMP molecules, cyclic-bis(3′→5′)-dimeric GMP, could be second messengers ubiquitous in bacteria, in which proteins containing GGDEF and EAL or HD-GYP domains were at the center of this regulation, being involved in the synthesis and degradation of c-di-GMP, respectively [1]. In the following years, different papers were published showing the central role of c-di-GMP orchestrating different signaling networks such as regulation of the flagellar rotor, bacterial motility such as twitching, exopolysaccharide synthesis, and regulation of bacterial biofilm formation. Nevertheless, c-di-GMP was first identified in 1987 as an allosteric activator of cellulose synthase in the cellulose-producing bacterium *Komagataeibacter* (*Gluconacetobacter) xylinus* [2]. It was the first c-di-GMP receptor described, and nowadays a huge range of different receptors have been identified, including RNA structures known as riboswitches. Therefore, a cyclic dinucleotide neglected in the microbiology area for 20 years emerged as a regulator of the bacterial cell lifestyle.

Recently, this research area has been under expansion, with the discoveries of new intracellular signaling cyclic dinucleotides (CDNs) in bacteria. In 2008, it was demonstrated that bacteria can produce not only c-di-GMP, but also c-di-AMP, cyclic-bis(3′→5′)-dimeric AMP, by an enzyme known as DisA that possess a DAC domain [3]. In 2012, a novel cyclic dinucleotide has been found to be a second bacterial messenger, cGAMP, cyclic guanosine (3′→5′) monophosphate-adenosine (3′→5′) monophosphate, synthesized by proteins containing SMODS domain such as the DncV protein [4,5]. At the moment, c-di-GMP, c-di-AMP and c-GAMP have been described as the main bacterial second messengers. Nevertheless, different classes of cyclic oligonucleotides, such as c-UAMP, c-di-UMP, c-UGM, c-CUMP, and c-AAGMP, have also been found in bacteria [2,3,5,6]. These molecules include not only di-purines but also hybrids of purine and pyrimidines and cyclic trinucleotides [6].

The cyclisation between two nucleotides of the most common bacterial CDNs involves the formation of a phosphodiester bond that links the C3’ of one pentose ring with the C5’ of another, resulting in a 3’-5’ cyclic dinucleotide (3′→5′). Despite their chemical similarities, there are specific enzymes involved in the synthesis and degradation of different CDNs. Furthermore, bacteria have different classes of CDN receptors that are specific to only one type of CDN. However, how the receptors differentiate one CDN from another is still unclear. Given the specificity of the receptor, since this is the molecule responsible for directly or indirectly regulating different bacterial phenotypes, changes in a single base of the CDN can lead to quite divergent biological responses, as described below.

Molecules of c-di-GMP generally coordinate the transition of a bacterium’s lifestyle, from a mobile single cell undergoing planktonic growth to a multicellular community in biofilm structures, a form of sessile growth. Regulation of these transitions are mediated by controlling the bacterial motility through the regulation of the flagellar rotor [7] and the twitching motility machinery [8]. Alternatively, in Streptomycetes, c-di-GMP regulates the transition from vegetative mycelial growth to the formation of reproductive aerial mycelium [9]. This dinucleotide is also involved in the regulation of bacterial adhesion, cell cycle progression and division, biofilm formation, quorum sensing [10], regulation of the type II (T2SS) [11], type III (T3SS) [12], and type VI (T6SS) [13] secretion system machineries, as well as the synthesis and secretion of virulence factors and pathogenesis [14,15,16,17,18]. Similarities in the roles of eukaryotic cyclins and bacterial c-di-GMP molecules have also been suggested. In eukaryotes, cyclins drive the cell cycle by regulating the activity of cyclin-dependent kinases and promoting the asymmetric replication of future cells [19]. 

Some similar biological roles have been observed between c-di-GMP and c-di-AMP molecules [20]. Nevertheless, few c-di-AMP synthesizing enzymes have thus far been studied, and the more well-known enzymes are more widely distributed and were better characterized in Gram-positive bacteria, but homologs can be found in several Gram-negative and a few archaeal lineages (Appendix A) [21,22]. Given its abundance and widespread distribution, c-di-GMP stands out as the main second messenger in bacteria. The c-di-AMP molecule regulates processes such as osmoprotection [23,24], cell-wall homeostasis [25], potassium ion channel expression and function [26], DNA repair to maintain genomic integrity [3], diverse gene expression [27,28], biofilm formation [29,30], sporulation[31], antibiotic resistance [32], and metabolism[33]. Another CDN, 3’-5’ cGAMP modulates chemotaxis, virulence and exoelectrogenesis (the use of insoluble extracellular terminal electron acceptors) [34]. 3’-5’ cGAMP and also c-UAMP activate the phospholipase activity of patatin-like lipase enzymes [6,35]. 

As described above, c-di-GMP molecule is synthesized by GGDEF domain-containing proteins [36,37,38] and degraded into pGpG or GMP by phosphodiesterase proteins containing HD-GYP or EAL domains, respectively. The released pGpG can then be degraded by an oligoribonuclease Orn [39,40]. A divergent GGDEF family enzyme synthesizes not only c-di-GMP but also 3’-5’ cGAMP molecules [41,42]. Bacteria synthesize 3’-5’ cGAMP mainly by the activity of proteins containing SMODS domains (Second Messenger Oligonucleotide or Dinucleotide Synthetase) [4,5], while eukaryotic cells synthesize cGAMP with a mixed 2′-5′ and 3′-5′ phosphodiester linkage (2′-3′ cGAMP) by GMP-AMP synthase (cGAS) enzymes [43,44,45]. cGAS are structurally similar to SMODS and they belong to cGAS/DncV-like nucleotidyltransferases (CD-NTases) enzymes superfamily [4,5]. The best characterized enzyme containing SMODS domain is DncV (a *Vibrio cholerae* dinucleotide cyclase), with its orthologues being able to synthesize different cyclic nucleotides, including 3’-5’ cGAMP, cUAMP, c-di-UMP, and cAAGMP, respectively produced by DncV, DncE, *Lp*CdnE02, and *Ec*CdnD02 proteins [6]. Bacterial CD-NTase enzymes also synthesize, as minor products, c-di-GMP, c-di-AMP, cUGMP, and cCUMP [6]. This diverse array of products synthesized by this group of enzymes is thought to be related to a low energetic barrier at the catalytic site for altering product specificity [6]. 

The c-di-AMP molecules are synthesized by di-adenylyl cyclase (DAC) proteins and hydrolyzed into pApA or AMP by specific phosphodiesterase (PDE) that contain DHH-DHHA1 or HD (His-Asp) domains [46,47]. *Listeria monocytogenes* encodes two PDEs: PdeA and PgpH [47]. Remarkably, the hydrolysis activity of PgpH is inhibited by the alarmone ppGpp, suggesting a crosstalk between c-di-AMP signaling and stringent response [47]. Three classes of DAC proteins have been identified, DisA, CdaA, and CdaS. All of them contain DGA and RHR motifs that are important in the catalysis [48,49]. 

Bacteria sense second messenger molecules in different ways and respond to them in different manners. The c-di-GMP, c-di-AMP and cGAMP molecules are sensed by both proteins and RNAs. Some examples of c-di-GMP effectors are mRNA riboswitches [50], transcription factors [9,51,52], and different classes of protein domains such as PilZ [53,54,55], degenerate GGDEF domains, degenerate EAL domains [56], and AAA+ ATPases domains [11,12,57] (see Table 1 for more details in the cyclic dinucleotide receptors section). When binding to its receptors, c-di-GMP can regulate bacterial physiology by altering gene transcription, protein expression and protein function, thus allowing the bacteria to respond to environmental stimuli at different speeds, be it a fast response, achieved by directly regulating protein activity, or a slower one, by regulating gene expression. Moreover, c-di-GMP can regulate the same physiological process in different ways, as seen for bacterial motility, when both flagellar gene expression and the flagellar motor function are regulated by c-di-GMP molecules [53,58]. Some examples of c-di-AMP effectors are proteins containing RCK domain [59], the universal stress protein (USP) domain [60], and PstA proteins [61]. In contrast, almost nothing is known about bacterial cGAMP receptors; one example is the CapV phospholipase from *V. cholerae* [35].

In the case of the 3’-5’ cGAMP receptors, genes are located adjacent to CD-NTase genes in the genome encode nucleotide receptors and act as effectors in biological conflicts, such as phospholipases, nucleases, and pore-forming agents [4,35]. The CD-NTase genes and the cyclic nucleotide receptors are generally found on mobile genetic elements, while genes encoding GGDEF domains are widespread in the chromosomes of different bacteria and c-di-GMP receptors do not show a tendency to be located close to genes encoding proteins containing GGDEF domains. 

CDNs are probably present in almost all kinds of bacteria and their impact on an organism’s physiology is probably determined by their concentration and the type of the second messenger. Interestingly, eukaryotic host cells evolved ways to sense some bacterial CDNs as a strategy to detect the presence of a pathogen and thus trigger a counterattack to avoid or fight invasion. The stimulator of interferon genes (STING) protein binds bacterial CDNs 3’-5’ cGAMP, 3’-5’ c-di-AMP and 3’-5’ c-di-GMP molecules, as well as eukaryotic 2’-3’ cGAMP molecules. Binding of STING to these cyclic dinucleotides activates expression of type I interferon in infected cells and initiates the innate immunity response for successful pathogen elimination [62]. On the other hand, cUAMP and cAAGMP are not recognized by STING proteins but are a ligand of mammalian CDN sensor reductase controlling NF-κB (RECON) [6]. Binding of c-di-AMP, cUAMP or cAAGMP to RECON inhibits its enzymatic activity, leading to increased activation of the proinflammatory transcription factor NF-κB, redirecting the cellular response toward an antibacterial reaction [6]. These new discoveries suggest that different classes of cyclic oligonucleotides molecules may have a larger role in bacterial signaling and pathogen recognition than previously thought. Consequently, STING could be a target for new drugs for the treatment of bacterial infections [1]. Additionally, it is noteworthy that STING is currently being explored as a candidate stimulant for anticancer immune activity [63]. 

In this review, we focus on recent advances in relation to the enzymes involved in the production of bacterial CDNs. We describe the conserved residues important to perform the catalysis of proteins containing GGDEF, SMODS, or DAC domains. Some bacterial CDN receptors that had their structure solved in complex with the ligand are also presented, with a focus on the residues involved in ligand recognition. We also highlight the conformation of the CDNs inside of the protein binding pocket. Surprisingly, different kinds of receptors bind CDNs with similar conformations. Additional observations, based on genomic data, suggest different CDN second messenger systems tend to coexist in many organisms showing the complexity and the importance of bacterial CDN signaling networks. We explore these resources and present an organization of our current knowledge on this expanding research topic.

## 2. GGDEF, SMODS, and DAC Domains Do Not Share Structural Similarities and Probably Perform the Nucleotide Cyclization Catalysis by Different Mechanisms

At the moment, three different classes of prokaryotic proteins are known to synthesize CDN molecules: (i) proteins containing GGDEF domains (Pfam family: PF00990); (ii) CD-NTases enzymes that have the catalytic domain known as SMODS (PF18144) [4]; and (iii) DAC proteins that have a catalytic domain called DAC domain (DisA_N domain, PF2457). 

Proteins containing GGDEF domains synthesize mainly 3’-5’ c-di-GMP (c-di-GMP) molecules, while proteins containing SMODS domain synthesize preferentially 3’-5’ cGAMP (cGAMP) molecules and proteins containing DAC domain synthesize mainly 3’-5’ c-di-AMP (c-di-AMP) molecules. Even though CDN molecules are mainly synthesized by prokaryotic cells, eukaryotic cells also synthesize CDNs such as 2’-3’ cGAMP by cGAS enzymes. These three classes of CDN synthetases do not share structural similarities, have different residues involved in substrate binding, and possess different catalytic mechanisms. Therefore, they are not homologs and probably evolved independently to catalyze analogous chemical reactions. 

Members of families within the CD-NTases superfamily, such as SMODS and cGAS, often do not share detectable primary sequence similarity but adopt a Pol-β-like nucleotidyl transferase fold, suggesting a common origin followed by divergent evolution [5,6,64,65]. cGAS and enzymes containing-SMODS domain use a single active site to sequentially form two separate phosphodiester bonds and release one cyclic nucleotide product. On the other hand, proteins containing DAC or GGDEF domains require homodimerization to perform catalysis. DACs adopt a unique, particular fold, while GGDEF domains are homologous to adenylyl/guanylyl cyclase catalytic domains and to the palm domain of DNA polymerases; see below [1,22]. 

Proteins containing GGDEF domains require an accessory domain that sense different signals to regulate the GGDEF homodimerization and consequently its enzymatic activity [66]. Each GGDEF domain binds one molecule of GTP and its dimerization positions the two GTP molecules in an antiparallel manner to enable their condensation into c-di-GMP with the release of two pyrophosphate molecules [67]. Therefore, proteins containing GGDEF domains are Bi Ter (two substrates, three products) enzymes and cannot be described by a Michaelis–Menten model [68,69]. A similar enzymatic mechanism seems to happen for proteins containing DAC domains. In the following sections, the structures of GGDEF, DAC and SMODS domains and the residues important to their catalysis are described in more detail. 

### 2.1. GGDEF Domain Structure and Catalysis

GGDEF structure and structural similarities with other protein domains. The GGDEF domain has an overall structure composed of a central five-stranded β sheet surrounded by five α helices [70] and one hairpin (Figure 1B,C). The GGDEF domain has structural similarities to three other catalytic domains: (a) the class III adenylate and guanylate cyclase catalytic domains (Guanylate_cyc, PF00211), (b) the GTP cyclohydrolase III (GCH_III, PF05165), and (c) the palm domain of family Y DNA polymerases, such as IMS domain-impB/mucB/samB family domain, PF00817) (Figure 1). All of these families have a similar structural core composed by a β-α-α-β-β-α-β-α-β topology (Figure 1B), which contains the Alpha-beta Plait topology (β-α-β-β-α-β), as defined by the CATH database [71]. The specific version of the Alpha-beta Plait topology embedded in this group is better known as the RNA Recognition Motif-like fold (RRM-like fold) and corresponds to the so-called “palm domain” shared by archaeo-eukaryotic primases, reverse transcriptases, viral RNA-dependent RNA polymerases and families A, B, and Y of DNA polymerases [72].

The catalytic domains of class III adenylyl cyclase (AC) and guanylyl cyclase (GC) are involved in the conversion of adenosine triphosphate (ATP) to 3′-5′ cyclic AMP (cAMP) and in the conversion of guanosine triphosphate (GTP) to 3′-5′ cyclic GMP (cGMP), respectively (Figure 1C) [73,74]. Class III AC and GC are well characterized: they are widely present in eukaryotic and prokaryotic cells and perform important function in many human tissues, being involved in signal transduction [66].

The GCH_III domain (GTP cyclohydrolase III) catalyzes the conversion of GTP to 2-amino-5-formylamino-6-ribosylamino-4(3*H*)-pyrimidinone 5′-phosphate (FAPy) [75]. GCH III catalyzes two modifications on the GTP molecule that involve two hydrolysis reactions, one at the base (a cyclohydrolase activity) and another in the phosphodiester bond (phosphotransferase reaction) that causes the release of a pyrophosphate molecule [75] (Figure 1C). Palm domains recognized by the IMS model of the Pfam database are the catalytic domains of DNA polymerases such as prokaryotic DNA polymerase IV and eukaryotic DNA polymerases eta and kappa [76]. All of them are Family Y DNA polymerases involved in DNA repair and exhibit error-prone behavior [77]. In these enzymes, the palm domain has deoxynucleotidyltransferase activity (Figure 1C). 

Given their conserved structural similarity to GGDEF domains, the class III adenylyl/guanylyl cyclases (AC/GC), GTP cyclohydrolase III, and the palm domain of DNA polymerases have been shown to be ancient homologous domains [78] that evolved from a common ancestor to perform different biological functions while preserving some core similarities such as: binding of nucleotides or deoxynucleotides and release pyrophosphate or phosphate during the enzymatic reaction course.

**Figure 1 molecules-25-02462-f001:**
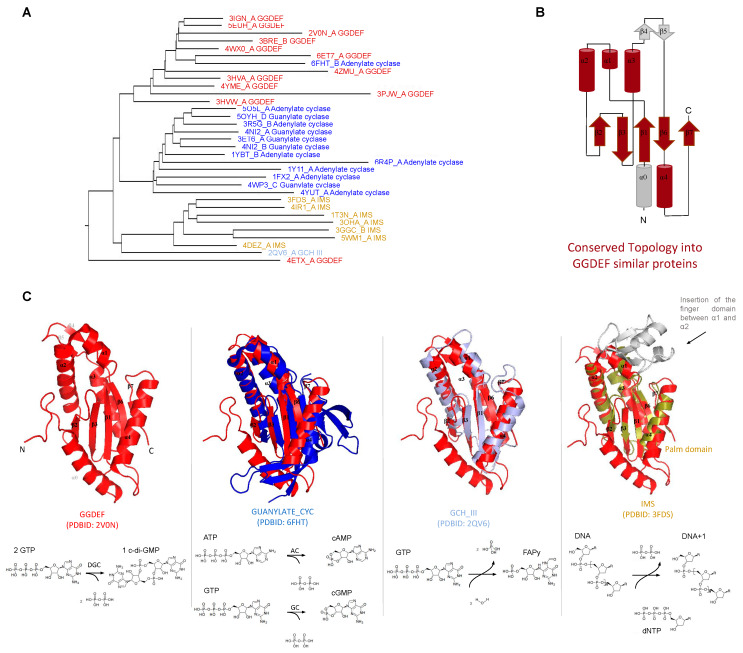
Structural similarities between GGDEF (**GG**[**D**/E][**E/**D]**F** conserved sequence motif) domains and adenylate/guanylate cyclase, GTP (guanosine triphosphate) cyclohydrolase III and RRM-like palm domain of DNA polymerases. (**A**) dendrogram showing structures similar to GGDEF domain made with the Dali server [79] (query: PleD PDBID: 2V0N). Each domain is colored with different colors and the PDBID_chain and the Pfam name are shown for each branch. The conserved fold found in most of these structures is shown in brown, panel (**B**), overlaid on the GGDEF domain of PleD topology. (**C**) structural superposition of GGDEF domain of PleD (PDBID: 2VON) with the other domains shown in the dendrogram (panel A), using the same colors to represent each domain. At the bottom of each structural alignment, the domain’s name and the PDBID code is shown, as well as the chemical reaction performed. DGC: diguanylate cyclase; AC: adenylate cyclase, GC: guanylate cyclase, and FAPy: 2-amino-5-formylamino-6-ribosylamino-4(3*H*)-pyrimidinone 5‘-phosphate.

Residues are important to GGDEF catalysis. The GGDEF domains are diguanylate cyclases (DGCs) that convert two molecules of GTP into one molecule of c-di-GMP. The GGDEF active site is thought to be assembled only when two GGDEF domains come together in such a manner that permits the nucleophilic attack of the 3’ OH groups on the α-phosphate groups of each GTP, leading to the synthesis of one molecule of c-di-GMP and two pyrophosphate molecules [70,80,81]. Therefore, DGCs are Bi Ter enzymes (two substrates, three products) as described above [68,69]. 

The catalytic activity of GGDEF domains is often regulated by input domains that precede the GGDEF domain, of which most are known or predicted to form dimers or heterodimers and to be sensor domains (Figure 2A). Isolated GGDEF domains have little or no detectable enzymatic activity [37,82] and require the dimerization of the input domain to assemble a catalytically competent GGDEF domain. Two hypotheses of the GGDEF activity regulation were reported. One of them suggests that the input domain binds its ligand and enhances the homodimerization and consequently the correct orientation of GGDEF domains to perform the catalysis. The other hypothesis suggests that the protein is a homodimer already and, when the input domain binds its ligand, it causes a reorientation of the GGDEF domains to a catalytically competent GGDEF dimer, or vice versa [67]. The signal transduction from the input domain to the GGDEF domain is predicted to be relayed by a S-helix (signaling-helix) that connects the two domains and forms a two-helical parallel coiled coil (stalk) in the dimer form of the protein [67,83]. Some proteins containing GGDEF domains possess a more complex activation mechanism and may involve formation of higher oligomers [67,84,85,86].

The GG(D/E)EF motif (glycine, glycine, aspartic or glutamic acid, and phenylalanine residues) is located in the loop between β2 and β3 (Figure 3A), in which the glutamic acid residue binds to the α-phosphate group of GTP molecule as well as coordinates one of the cations located in the binding site (Figure 3B). In the case of the PleD GGDEF domain, two magnesium cations are located in the binding site and are coordinated by E370 (from the GG(D/E)EF motif), D327, and the main chain of I328. The PleD residues D344 and N335 bind the guanosine base of the substrate, while the side chains of E370, K442, R446 and the main chains of F330, F331, and K332 bind the phosphate moieties of the GTP molecule (Figure 3B) [80]. The GG(D/E)EF consensus sequence and most of the residues important to catalysis are very well conserved within GGDEF family members (Figure 3A). This includes the D327, N335 and D344 residues, which have been reported to be essential to GGDEF domain activity [70,87].

**Figure 2 molecules-25-02462-f002:**
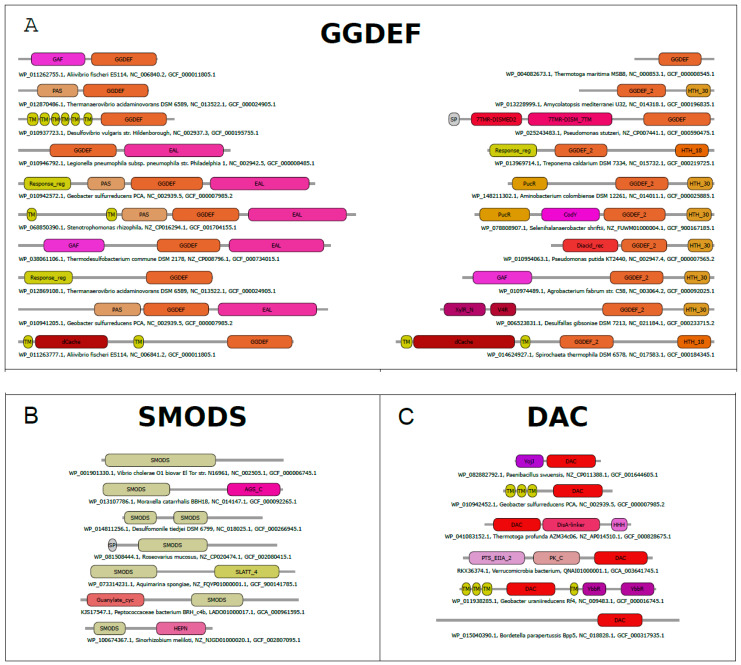
Domain architectures of proteins containing GGDEF, SMODS (Second Messenger Oligonucleotide or Dinucleotide Synthetase), and DAC (di-adenylyl cyclase) domains. The most frequent domain architectures of proteins containing GGDEF (**A**), SMODS (**B**) and DAC (**C**) domains. The analysis was done using a non-redundant dataset (<80% identity) of protein sequences built from sequences retrieved from the NCBI protein database [88]. The names of the domains are based on the Pfam database [89].

A subclass of GGDEF domain, called Hybrid promiscuous (Hypr) GGDEF enzymes, synthesizes predominantly cGAMP molecules, but also synthesizes c-di-AMP and c-di-GMP molecules [41]. The change in the substrate specificity seems to be related with the substitution of an aspartate (D344 of PleD located at the α2) by a serine, exactly the residue that binds the guanine base of the GTP in PleD.

Some GGDEF domains have diverged from the canonical GG(D/E)EF amino acid sequence motif and are described as degenerate GGDEF domains, due to the loss of their catalytic activity. These degenerate GGDEF domains can evolve to possess different biological functions, and two examples have been described in the literature: 1-a degenerate GGDEF domain that is a sensor domain and binds GTP to activate the phosphodiesterase activity in the neighboring EAL domain of the *Caulobacter crescentus* CC3396 protein [38]; and 2-a degenerate GGDEF domain of the *Bacillus subtilis* YybT protein that has unexpected ATPase activity [92].

Allosteric inhibition in proteins containing GGDEF domain. The DGC activity of proteins containing GGDEF domain are inhibited by an allosteric noncompetitive product inhibition. A GGDEF dimer contains two symmetrical allosteric sites (I and I’ sites), in which each allosteric site binds a c-di-GMP dimer (c-di-GMP)_2_ (Figure 3B). Both sites are formed by four residues, three of them from one GGDEF molecule, the RxxD motif (R359 and D362 of PleD) and an arginine (R390 of PleD), and the fourth residue is an arginine from the adjacent GGDEF molecule (R313 of PleD). The two (c-di-GMP)_2_ dimers are expected to crosslink allosteric sites on opposite GGDEF domains, resulting in their immobilization in an inactive orientation [70,80,87,93] (Figure 3B). The RxxD motif and the positively charged residue (R390 in the case of PleD) are conserved in GGDEF members (Figure 3A).

### 2.2. SMODS Domain Structure and Catalysis 

SMODS structure and structural similarities with other protein domains. The *Vibrio cholerae* dinucleotide cyclase (DncV, the gene product of VC0179) has two domains, a SMODS domain located at its N-terminus and an **A**denylyl/**G**uanylyl and sMODS C-terminal sensor domain (AGS-C) [4] at its C-terminus [94]. The first 23 residues of the protein are located in the AGS-C domain, which presents a mainly α-helical structure. The SMODS domain has two β-sheets connected by one β-strand (β3). It also has six α-helices that do not make part of the interface between the two domains. The two β-sheets are composed by the strands: β2-β3-β7-β8-β9 and β3-β6-β5-β (Figure 4A). The substrate binding site is located in the interface between the two domains, in which the SMODS β-sheets make close contacts with the substrate (Figure 5A). Proteins containing SMODS domains are also found associated with other domains (Figure 2B) and, in rare cases, can be found in proteins containing two enzymatic domains: a SMODS and a class III AC/GC catalytic domain, both domains related with synthesis of cyclic nucleotide second messengers. 

DncV have structural similarities with proteins belonging to the nucleotidyltransferase (NTase) fold, a highly diverse superfamily of proteins (Figure 4C,E) [95]. NTase fold structure is characterized by the presence of a minimal conserved core of a mixed β-sheet flanked by α-helices (α1-β1-α2-β2-α3-β3-α4) that correspond to α3-β2-α8-β3-α9-β6 in DncV protein, missing the α4 element (Figure 4A,B). This common core is usually decorated by various additional structural elements depending on the family. The NTase fold core is present in the DncV SMODS domain and various insertions are observed (Figure 4A,B). Members of NTases superfamily contain three conserved motifs located at the active site: (i) hG[G/S] located at α2; (ii) [D/E]h[D/E]h (h indicates a hydrophobic amino acid) located at β2, and (iii) h[D/E]h located at β3 [95]. These three conserved motifs in DncV protein correspond to: (i) G_113_-S_114_ (located at α8); (ii) D_131_-I_132_-D_133_ (located at β3); and (iii) N_193_ (located at β6). Two of these motifs are conserved in proteins containing SMODS domains, the hG[G/S] and [D/E]h[D/E]h motifs (Figure 4A).

More specifically, as revealed by Dali searches [79], the DncV structure shares structural similarities with eight members of the Nucleotidyltransferase superfamily (NTS): (**i**) proteins containing DZF domain (domain associated with Zinc Fingers, and PFAM model PF07528) such as nuclear factor NF90 and NF45; (**ii**) the catalytic domain of Poly(A) polymerase, PAP, (PAP_central domain, PF04928) [96]; (**iii**) the catalytic domain of eukaryotic cGAS enzymes (Mab-21 domain, PF03281); (**iv**) the D1 and D2 domains of the U3 small nucleolar RNA-associated protein 22 (Utp22) [97], in which the D1 domain matches the PFAM model named nucleolar RNA-associated proteins domain (Nrap, PF03813) and the D2 domain matches the Nrap protein PAP/OAS-like model (Nrap_D2, PF17403) [98]; (**v**) the OAS1-like domain of the dsRNA-activated oligoadenylate synthase (OAS) protein, which is described in the PFAM database as two domains: an N-terminal domain matching the NTP_transf_2 model (PF01909), also known as N-lobe of human OAS3 pseudoenzymatic domain DI (hOAS3.DI), and a C-terminal OAS1_C domain (PF10421), also known as C-lobe of hOAS3.DI [99]; (**vi**) the catalytic and central domains of Poly(A) RNA polymerase protein 2 (TRF4 gene), where the catalytic domain matches the NTP_transf_2 (PF01909) and the central domain matches the PAP_assoc model (Cid1 family poly A polymerase, PF03828) [100,101]; (**vii**) proteins matching the GrpB domain (PF04229) [102], this domain is found in uncharacterized proteins such as EF_0920 from *Enterococcus faecalis*; and (**viii**) the palm subdomain (DNA_pol_B_palm, PF14792) [103] of the DNA polymerase μ (Pol μ) from the family X [104] that includes DNA polymerase β, γ, and μ [105] (Figure 4E). 

All of these proteins share not only the NTase fold core, but also some secondary structures from the AGS-C domain (Figure 4D) suggesting that the domain interface is conserved in these families. It is worth mentioning that the AGS-C domain shares structural similarities with domains that are commonly associated with the catalytic domain of members of NTase, such as DZF C-terminal domain, OAS1_C, PAP_assoc, and Nrap_D2 (Figure 4E). 

DZF domains form dimers and heterodimers and are found in proteins involved in gene expression and RNA metabolism such as NF90 that forms a complex with NF45 and regulates genes expression [106]. Poly(A) polymerase (PAP) is involved in eukaryotic mRNA processing by its polyadenylation at the end of transcription process, so PAP incorporates ATP at the 3’ end of Mrna [107]. In metazoans, the cGAS enzyme, which has a Mab_21 domain, binds cytoplasmatic double-stranded DNA (dsDNA) to activate synthesis of 2’-3’ cGAMP molecules and initiate host innate immune responses. Endogenous or exogenous dsDNA in the cytoplasm, which could be from damaged mitochondria or from an invasion of pathogenic bacteria or viruses, indicates major danger to eukaryotic cells. The cytosolic accumulation of 2’-3’ cGAMP activates type-1 mediated stress-responses via STING and regulates autoimmunity in human cells [108]. Human dsRNA-activated oligoadenylate synthase (OAS), which matches both NTP_tranf_2 and OAS1_C PFAM models, is a mammalian dsRNA sensor, which is increased during pathogen infections, and activates the synthesis of a second messenger 2’-5’-linked RNA molecules to cause RNA decay [109,110]. 

TRF4 protein, which also contains a region most similar to the NTP_Tranf_2 model, makes part of a polyadenylation TRAMP complex that recognizes aberrant eukaryotic RNAs and target them for degradation [111]. Members of DNA polymerase family X, which have a “palm” subdomain, play essential roles in the base-excision repair mechanism, a process that repairs cell DNA base damage, being responsible for DNA synthesis and 5’-deoxyribose-phosphate (dRP) removal (dRP lyase activity). These enzymes can be also involved in other DNA repair processes such as non-homologous end-joining and lesion bypass [105,112,113]. Utp22, which has a Nrap_D2 domain, forms a complex with Rrp7 and they are present in early precursors of small ribosomal subunit. Utp22 is a structural building block and apparently lacks any enzymatic activity [97]. 

Among all described enzymes, cGAS enzymes are the only ones that share functional similarities with the prokaryotic DncV enzymes. While DncV synthesizes 3′-5′ cGAMP molecules, cGAS enzymes synthesize 2’-3′ cGAMP. Most of the other enzymes with structural similarity with the DncV SMODS domain have functions related to DNA or RNA processing.

Active site of SMODS domains—DncV synthesizes preferentially cGAMP, but also produces c-di-AMP and c-di-GMP molecules. DncV regulates the expression of more than 80 genes in *Vibrio cholerae* and its DGC activity is inhibited by folate-like molecules in vitro [5,94,114]. Orthologs of DncV are found in Gram-negative and Gram-positive bacterial species and the residues involved in ligand binding and to folate-like molecule binding are conserved among them [94] (Figure 5A). The active site is located in the interface between SMODS and AGS-C domains, while the folate-like molecule binds at the opposite of the substrate-binding pocket at the flat side of the protein (Figure 5A). The folate-like molecule binds mainly at the SMODS domain and makes few interactions with the linker between the two domains (Figure 5). The inhibitory site of the DncV, which binds folate-like molecules, such as 5-methyltetrahydrofolate diglutamate (5MTHGLU2), is formed by side chains of Arg36, Arg40, Arg44, Arg108, Trp110, Gln116, Tyr137, Phe204, and Asp260, by the main chain of Phe109, Thr111, and Leu240 and by a hydrophobic pocket formed mostly by the side chains of Leu240 and Val245 (Figure 5B). These residues are not conserved in members of SMODS (Figure 5A). 

The active site of DncV is built by nine residues: Ser114, Tyr117, Asp131, Asp133, Arg182, Ser259, Lys287, Ser301, and Asp348. Asp348 and Ser259 interact with the guanine and adenine bases of the substrate, respectively. Tyr117 and Asp133 are involved in the interaction with the ribose of the guanine and the adenine nucleotides, respectively. Arg182 binds the β and γ phosphate groups of the ATP. Tyr117, Ser114, Lys287, and Ser301 interact with the β and γ phosphate groups of GTP. The magnesium ion is coordinated by Asp131, Asp133, and the α, β and γ-groups of the GTP ligand (Figure 5B). These two aspartic residues belong to the Dh(D/E) motif conserved in members of NTS, as described above, and are key residues in (2’-5’) oligoadenylate synthetase (OAS1) and poly(A) polymerase activities [115].

**Figure 5 molecules-25-02462-f005:**
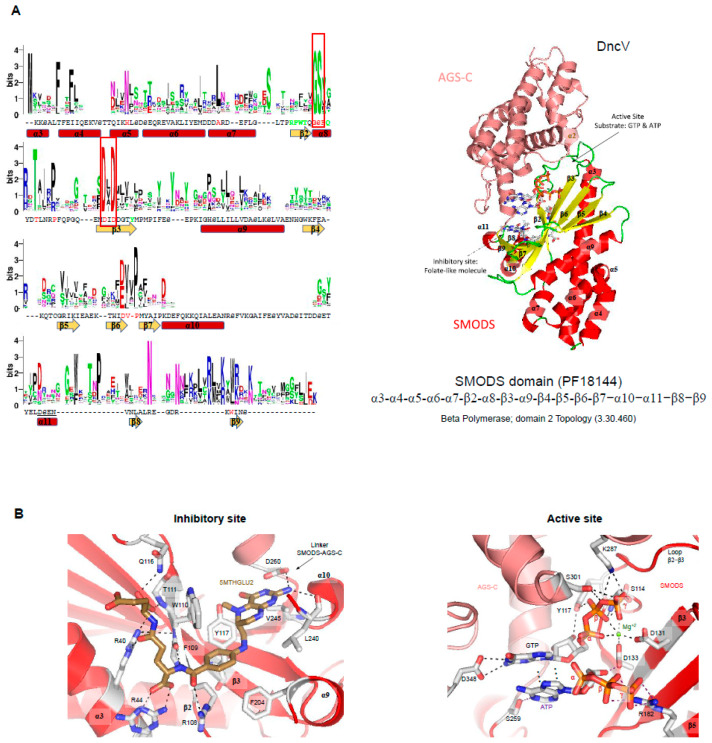
Conserved sequences within SMODS members and their substrate and inhibitor binding pockets. (**A**) residue frequency in SMODS proteins. Forty-five sequences were used from the Pfam database to create a multiple sequence alignment of the SMODS domain of different proteins using ClustalW [116], and the sequence logo was done using the WebLogo server [90]. The sequence shown below the logo and the secondary structure elements belong to the *Vibrio cholerae* dinucleotide cyclase DncV (PDBID: 4U03). Residues colored in green are involved in the interaction between the DncV SMODS domain with the folate-like inhibitor, 5-methyltetrahydrofolate diglutamate (5MTHFGLU2) molecule (residues Arg36, Arg40, Arg44, and Asp260, be located at the AGS-C domain, are not shown). Residues located at the SMODS domain involved in the catalytic activity are colored in red (residues Ser259, Lys287, Ser301, and Asp348, located at the AGS-C domain, are not shown). The red boxes contain the G(G/S) and Dx(D/E) motifs found in members of NTS. The structure shown in the right belongs to the DncV protein (PDBID: 4U03), in which the AGS-C is colored in salmon and the SMODS domain is colored by secondary structure (β-strands in yellow and α-helices in red). The SMODS domain topology is shown below the structure, and the CATH topology name and code are also shown [91]; (**B**) interaction network between the DncV binding pocket (active site) and its inhibitory site with substrate and inhibitor molecules, respectively. The substrates GTP and ATP are found bound at the active site, while the folate-like molecule (5MTHGLU2) binds at the allosteric site, inactivating the protein. The residues interacting with substrate and inhibitor molecules are shown as sticks and colored by element: the inhibitor carbons are colored brown, the magnesium ion is shown as a green sphere, and gray dotted lines represent hydrogen bonds.

### 2.3. DAC Domain Structure and Catalysis

DAC structure and structural similarities with other protein domains. c-di-AMP is synthesized by DAC enzymes that convert two molecules of ATP into one c-di-AMP and two pyrophosphate molecules. In the case of DisA, the Rv3586 protein from *Mycobacterium tuberculosis*, the synthesis of c-di-AMP is made using ATP or ADP [117]. Production of c-di-AMP has been described as essential for the growth of some Gram-positive bacteria due to it being involved in crucial cellular activities, such as cell wall metabolism, maintenance of DNA integrity, ion transport, cell division, and cell size control [22,118,119,120]. *Bacillus subtilis* encodes three DAC enzymes, DisA, CdaA, and CdaS. Two of them, DisA and CdaA, are constitutively expressed during vegetative growth while CdaS is required for efficient germination of spores. Other Gram-positive bacteria encode only one DAC protein that is essential for their growth, as observed in *Listeria monocytogenes, Streptococcus pneumoniae*, and *Staphylococcus aureus*, thus making this enzyme a likely target for constructing new inhibitors that may serve as antibiotics for pathogenic Gram-positive bacteria.

At the moment, three structures of proteins containing DAC domains have been solved: DisA from *Thermotoga maritima* (PDBID 3C1Y) [3], DisA (named in the UniProt database as DacB) from *Bacillus cereus* (PDBID 2FB5) [121]; and DacA (CdaA-APO Y187A Mutant) from *L. monocytogenes* (PDBID 6HVN) [122]. As described before, no structural similarities with other domain were detected so far. The overall DAC domain structure exhibits a globular α/β fold with a slightly twisted central β-sheet, made up of seven mixed-parallel and antiparallel β-strands (β1-β7) surrounded by five α-helices (α1-α5), in which the N-terminal helix (α1) can be split in two parts (α1’ and α1). Like GGDEF domains, two DAC domains must be correctly oriented to allow the conversion of two ATP molecules into one c-di-AMP molecule and two pyrophosphates. Therefore, DAC domains are also Bi Ter enzymes (two substrates, three products). The regulation of the catalytic activity of DAC domains may be regulated by input domains (Figure 2C), and in the case of DisA from *T. maritima* (PDB code 3C23, 3C1Z, and 3C1Y) [3], the protein forms a homo-octamer and the DAC domains are oriented in a such way that two DAC domains are oriented face to face to allow the catalysis. Therefore, in each DisA homo-octamer, there are four potential catalytic sites. Linear DNA or DNA ends do not affect the protein activity but branched nucleic acids (such as in Holliday junctions) strongly suppress the DAC activity of DisA by binding to its C-terminal domain [3].

Active site of DAC domains—In the case of the DAC domain of CdaA from *Listeria monocytogenes* (PDBID 4RV7), the ATP ligand is located in a well-defined cavity made up by the N-terminus of α4, loop β5-β6, loop β4-α4, and loop α3-β3 (Figure 6C), in which many conserved residues of DAC domains are located: the GALI motif, GxRHRxA motif, an absolutely conserved serine, DGAhh motif (h is a hydrophobic residue), and (V/I)SEE motif (Figure 6A). 

The active site of CdaA DAC domain is thought to be built by 10 residues: the side chains of Leu31, Asp71, Thr102, Arg103, His104, Ser122, and Glu124 and the main chain of Leu88, Gly101, and Glu123 (Figure 6C). Leu31 belongs to the GALI motif, the Asp71 belongs to the DGAhhh motif, and Thr102, Arg103, and His104 belong to GxRHRxA motif. The main chain of Leu88 and the side chain of L31 interact with the adenine base, Asp71 binds the ribose, and Thr102, Arg103, His104, Ser122, Gly101, and Glu123 bind the phosphate groups of the ATP molecule. The Glu124 coordinates one magnesium cation that binds the α and β phosphate groups of the ATP molecule (Figure 6C) [49]. CdaA from *L. monocytogenes* is active in the presence of Mn^2+^ or Co^2+^ but inactive in the presence of Mg^2+^ ions [122]. However, in the case of DisA from *M. tuberculosis* [117] and *T. maritima*, the enzymes are active in the presence of Mg^2+^ ions. 

## 3. Cyclic Dinucleotide Receptors

Different classes of CDN receptors have been described and are involved in the regulation of a broad range of bacterial behaviors, while, in eukaryotic cells, they are associated with the activation of innate immune response through interactions with STING proteins. In order to analyze the residues involved in CDN ligand and to compare the ligand structure inside of the protein pocket, we focused on the CDN receptors with three-dimensional structures solved and deposited in the Protein Data Bank (PDB) (Table 1).

C-di-GMP receptors are the most studied CDN receptors, probably because it was the first CDN identified as a second bacterial messenger. Therefore, most of the receptors with the structure solved in complex with the ligand analyzed in this review are c-di-GMP receptors. Examples of these are STING proteins and proteins containing PilZ domains, degenerate EAL domains, allosteric site of DGC enzymes containing GGDEF domains, and RNA structures such as c-di-GMP I and c-di-GMP II riboswitches. Other examples of c-di-GMP receptors are the C-terminal domain of the master regulator of *Streptomyces* cell development BldD, REC domain of the transcriptional regulator VspT protein, N-terminal domain of the ATPase of the Type II secretion system MshE protein (T2SSE_N domain), MerR domain of BrlR protein, AAA+ ATPase domain (Sigma54_activat, PF00158) of the transcriptional regulator FleQ protein, and CA domain (HATPase_c, PF02518) of the cell cycle kinase CckA protein (Table 1). 

Different classes of c-di-AMP receptors have also been described, such as proteins containing STING domains, Aldo-keto reductase domain of oxidoreductase RECON proteins, cyclic-di-AMP receptor domain of PII-like signal transduction protein (PtsA), pyruvate carboxylase domain (HMGL-like domain, PF00682) of *L. monocytogenes* pyruvate carboxylase (LmPC) or *Lactococcus lactis* pyruvate carboxylase (LIPC), TrkA_C domain of potassium transporter A (KtrA), CBS domain of OpuC carnitine transporter, and RNA structures such as ydaO-yuaA riboswitches (Table 1). For cGAMP, the receptors analyzed in this review are only STING proteins and c-di-GMP I riboswitches.

The function of each CDN receptor, as well as the residues involved in ligand binding, are described in more detail in Table 1. It is notable that most of CDN receptors are specific to their ligands, with the exception of receptors involved in mammalian cell innate immunity, such as STING that interact with different CDNs such as c-di-GMP, c-di-AMP and 3’-5’ and 2’-3’ cGAMP molecules (Figure 7B). Interestingly, even though the CDNs are chemically different, the STING binding pockets for each kind of CDN are very similar and the residues involved in ligand binding for each CDN are almost the same (Figure 7A,C). This suggests that STING adjusts the ligand binding site for each CDN by placing or removing water or magnesium molecules.

STING proteins are localized on the endoplasmic reticulum membrane of eukaryotic cells and are CDN sensors that, when bound, regulate the induction of type I interferons (IFN-α and IFN-β), thus eliciting the intracellular signals of the invasion by bacteria and/or viruses, and activating the innate immune response to attack the pathogen. STING proteins can directly sense the pathogen invasion by interaction with bacterial CDNs (3’-5’ c-di-GMP, 3’-5’ c-di-AMP or 3’-5’ cGAMP) or indirectly by binding to eukaryotic 2’-3’ cGAMP through its C-terminal domain (TMEM173, PF15009). It is controversial whether STING binds 2′-5′ cGAMP preferentially in relation to other CDNs, or binds all of them with the same affinity [114]. 

Other important c-di-GMP receptors are proteins containing PilZ domains (PF07238). PilZ domains regulate twitching and swarming motility via the flagellar regulator YcgR protein [7], but proteins containing different domain architectures are related with other functions, such as the regulation of the synthesis of cellulose by BcsA in *Rhodobacter sphaeroides*, or chemotaxis by MapZ protein and alginate secretion by Alg44 to promote biofilm formation in *Pseudomonas aeruginosa* (Figure 7E and Table 1). PilZ domain is found associated with different domains that could be sensor domains, such as GAF, Cache, and PAS domains, and catalytic domains such as GGDEF, EAL, and Peptidase_S8. Therefore, proteins containing PilZ domain could be classified based on their domain architecture and function in different paralogous families [124,125]. The Pfam database describes 221 different domain architectures containing PilZ domains [89], showing the diversity of signaling networks in which c-di-GMP can be involved and have not yet been explored. 

It is interesting that two proteins containing PilZ domains that are c-di-GMP receptors have been found to be involved in the production of different exopolysaccharides to produce bacterial biofilms: cellulose and alginate. Moreover, another c-di-GMP receptor is also involved in exopolysaccharide production, the PelD of *Pseudomonas aeruginosa* that regulates the synthesis of the Pel exopolysaccharide (Table 1).

PilZ proteins interact with c-di-GMP by two conserved sequence motifs: RxxxR and DxSxxG motifs (Figure 7D). In the RxxxR motif located in a loop at the N-terminal part of the PilZ domain, each arginine is interacting with the phosphate group and the base of the ligand. In the case of DxSxxG motif, the aspartic acid, serine and glycine residues bind the base and the pentose ring of the c-di-GMP molecule (Figure 7G). Other residues not conserved within members of the PilZ family are also involved in ligand binding and some of them are located at the β-strand 7 of the PilZ protein (Figure 7D,F,G). Some PilZ proteins lost their canonical residues to bind c-di-GMP and are not c-di-GMP receptors anymore but may work as protein–protein adaptors, as happens with the complex FimX-PilZ-PilB that regulates the twitching motility in *Xanthomonas citri* [8]. This ternary complex is an example of a full set of "degenerate" GGDEF, EAL, and PilZ domains, in which GGDEF does not synthesize c-di-GMP, PilZ does not bind c-di-GMP, and the EAL domain does not cleave c-di-GMP but kept the ability to bind it [8].

Degenerate EALs proteins lost their ability to cleave c-di-GMP to pGpG, and some of them still bind c-di-GMP molecules but do not cleave them changing its function from enzyme to a CDN receptor. The residues involved in c-di-GMP interaction are described in Table 1 and Figure 7H,I. The loss of the EAL domain catalytic function seems to be related with a change in the residues important for the coordination of a magnesium cation (Figure 7H,I).

In *Xanthomonas citri*, *Xanthomonas campestris*, and *Pseudomonas aeruginosa*, FimX proteins regulate twitching motility by sensing c-di-GMP levels via interaction with degenerate EAL domain and regulates type IV pilus machinery [8]. LapD from *Pseudomonas fluorescens* is a transmembrane protein that binds c-di-GMP through its C-terminal degenerate EAL domain to prevent cleavage of the surface adhesin LapA and therefore activates biofilm formation [126].

Different classes of RNA riboswitches sense different kinds of CDNs (Table 1). Riboswitches are structured RNAs located in the 5’-untranslated regions of mRNAs and some can sense CDNs molecules to change its structure to regulate expression of downstream genes that could be involved with virulence, motility, biofilm formation, cell wall metabolism, synthesis and transport of osmoprotectants, sporulation, and other important biological processes [127,128].

There are three distinct classes of riboswitches that bind specific CDNs and have had their structures solved in complex with their ligand and deposited in the Protein Data Bank: c-di-GMP I riboswitch (RF01051), c-di-GMP II riboswitch (RF01786), and c-di-AMP riboswitch (ydaO-yuaA riboswitch, RF00379). C-di-GMP I riboswitch and c-di-GMP II riboswitch bind c-di-GMP molecules while c-di-AMP riboswitch binds c-di-AMP molecules [28,129,130]. The c-di-GMP I riboswitch was originally annotated as a conserved RNA-like structure of Genes Related to the Environment, Membranes and Motility (GEMM motif) and later another c-di-GMP riboswitch class was identified, the c-di-GMP II riboswitch. They have the same function but do not share any sequence motif or structural similarities. The c-di-AMP riboswitch is one of the most common riboswitches in various bacterial species and is found in the vicinity of genes related to cell wall metabolism, sporulation in Gram-positive bacteria, and other important biological processes [127,128]. These structures reveal that the RNAs use different ways to bind CDNs.

The TetR-like transcriptional factor, DarR, from *Mycobacterium smegmatis* was the first c-di-AMP receptor discovered [27], where c-di-AMP stimulate the DNA binding activity of this protein. DarR is a repressor that negatively regulates the expression of its target genes [27]. Another protein that interacts with c-di-AMP by a poorly understood mechanism is KdpD/KdpE that controls the potassium uptake in situations where the potassium concentrations are extremely low and other uptake systems wouldn’t be enough to give the cell all potassium it requires. In *Escherichia coli*, there are three systems responsible for potassium uptake, namely, Trk, Kdp, and Kup. In the case of Trk system, four genes are constitutively expressed and TrkA is the predominant potassium transporter at neutral pH. The Kdp-ATPase system is induced at low potassium concentrations and under conditions of osmotic stress. The Kup, formerly TrkD, is activated when TrkA and Kdp activities are not sufficient [131,132,133]. In *Bacillus subtilis*, a novel high-affinity transporter KimA (formerly YdaO) has recently been characterized and the expression of KimA and KtrAB is negatively regulated by c-di-AMP riboswitches [28]. When the concentration of potassium is high in the cell, the concentration of c-di-AMP increases inhibiting potassium uptake by two ways, by binding to c-di-AMP riboswitches that will avoid the expression of proteins involved in transport, and by direct interactions with regulatory subunits of KtrAB and KtrCD causing the inhibition of potassium transport [134]. A similar process seems to happen in *Staphylococcus aureus*, where c-di-AMP binds to the KtrA protein and to the universal stress protein (USP) domain of the KdpD sensor kinase inhibiting the expression of Kdp potassium transporter components. In this manner, c-di-AMP appears to be a negative regulator of potassium uptake in different Gram-positive bacteria [60,134].

One of the most well understood receptors for c-di-AMP is KtrA, which binds c-di-AMP through its C-terminal domain (RCK_C or TrkA_C) to cause inactivation of the KtrA function (Table 1). c-di-AMP binds to the interface of the KtrA homodimer, and the residues involved in the ligand interaction are described in Table 1. Another c-di-AMP receptor is the c-di-AMP receptor domain (PF06153) of the PII-like signal transduction protein, PstA. PstA is a homotrimer and, in each protein interface, one c-di-AMP molecule is bound. The residues involved in ligand binding in PstA are also described in Table 1. 

c-di-AMP is also related with negative control of aspartate and pyruvate pools in *Lactococcus lactis* by a pyruvate carboxylase, LlPC protein, and *Listeria monocytogens* pyruvate carboxylase, LmPC protein, respectively. In both cases, c-di-AMP binds to the pyruvate carboxylase domain (HMGL-like domain in the Pfam) (Table 1). LIPC forms a tetramer and each c-di-AMP molecule binds the protein dimer interface at the carboxyltransferase (CT) domain in a binding site pocket containing residues that are poorly conserved among pyruvate carboxylases [135].

The huge repertoire of CDN receptors demonstrates the complexity of CDN signaling networks in bacteria. Additionally, CDNs may regulate different bacterial behaviors at different speeds through regulation of gene transcription by transcriptional factors, protein translation by riboswitches, and directly by regulating the function of different classes of protein. 

**Table 1 molecules-25-02462-t001:** List of the bacterial c-di-GMP, c-di-AMP, cGAMP, and eukaryotic cGAMP receptors that had their structure solved in complex with their ligand and deposited in the Protein Data Bank (PDB). The Pfam/Rfam and, in some cases, the InterPro domain is described. The residues involved in ligand binding are also described for a representative of each receptor.

Receptor Class (Pfam/Rfam)	Organism (PDBID)	Receptor Function	Ligand Binding Site	Ref.
3′-5′ c-di-GMP				
**STING** (TMEM173, PF15009)	*Homo sapiens* (4EF4, 4EMT, 6RM0, 6S86, 4F9G, 4F5D, 4F5Y)	Members of Transmembrane Protein 173 (TMEM173) family, also known as Stimulator of Interferon Genes (STING), are an important component of the immune system. STING proteins are responsible for regulating the induction of type I interferon via activation of INF-β gene transcription. Human STING (carrying the more common R232 allele) binds eukaryotic 2’-3′ cGAMP with high affinity compared with bacterial CDNs such as c-di-GMP, c-di-AMP, and 3′-5′ cGAMP [136]. Nevertheless, it is controversial whether STING binds 2′-5′ cGAMP preferentially since others STINGs binds CDNs with the same affinity [114].	STING proteins interact with c-di-GMP at the protein dimer interface in a perfectly symmetrical manner increasing the homodimer stability. This binding involves a hydrophilic core, that in the human STING (PDB 4F5D) corresponds to, S162, G166, Y167, R238, Y240, S241, N242, E260, T267, and the presence of two Mg^2+^ ions and two water molecules (Figure 7A–C). STING proteins bind monomers of c-di-GMP that are stabilized in the protein pocket at *intermediate* or *closed* conformations, Figure 8.	[137,138,139,140,141,142]
	*Sus scrofa* (6A04)			[143]
	*N. vectensis* (5CFL, 5CFP)			[144]
**c-di-GMP I Riboswitch** (RF01051)	*V. cholerae* (3MXH, 3MUT, 3MUR, 3MUM, 3IRW)	c-di-GMP Riboswitches, also known as GEMM (Genes for the Environment, Membranes and Motility), are structured RNAs located in the 5′-untranslated regions of mRNAs that sense c-di-GMP molecules to regulate expression of downstream genes that could be involved with virulence, motility and biofilm formation. Despite having the same function, the c-di-GMP I Riboswitch and c-di-GMP II Riboswitch do not share any sequence motifs or structural features.	GEMM Riboswitches interacts with c-di-GMP by an uncharacterized motif with high affinity, at the picomolar range, compared to c-di-GMP protein receptors, with nanomolar to micromolar affinities. In the case of c-di-GMP I Riboswitch (PDB 3IRW) the nucleotides involved in ligand binding are: G14, C15, A16, C17, A18, G19, G21, C46, A47, A48, A49, G50. c-di-GMP II riboswitch (PDBID 3Q3Z) binds to c-di-GMP through the nucleotides: A13, A14, U37, G39, U60, A61,C68, A69, A70, C71, C72, G73, and A74. Riboswitches can recognize the guanine base of the ligand in different ways. The ligand was found as *closed monomers*, Figure 8.	[145,146]
	*Geobacter* (4YB0)			[147]
	*E. coli* (3IWN)			[148]
**c-di-GMP II Riboswitch** (RF01786)	*C. acetobutylicum* (3Q3Z)			[146]
**PilZ domain** (PF07238)	*V. cholerae* (2RDE)	VCA0042 is an important protein for the efficient infection of mice by *V. cholerae*. This PilZ-containing protein senses the bacterial second messenger c-di-GMP and controls virulence factors.	This PilZ domain interacts with monomeric c-di-GMP via two main sequence motifs: RxxxR and DxSxxG motifs (PDBID: 2RDE), Figure 7D, E. The ligand was found as *intermediate monomers*, Figure 8.	[124]
	*R. sphaeroides* (5EIY, 5EJ1, 5EJZ, 4P00, 4P02)	BcsA, Bacterial cellulose synthase A, is a component of a protein complex that synthesizes and translocates cellulose across the inner membrane. The binding of c-di-GMP to a complex BscA and BcsB releases the enzyme from an autoinhibited state, generating a constitutively active cellulose synthase.	Most PilZ domains interact with dimeric c-di-GMP, in which one molecule interacts with two main sequence motifs on the β-barrel surface, DxSxxG and RxxxR motifs (PDBI: 5EIY, 5EJ1, 5EJZ, 4P00, 4P02, 5Y6F, 5Y6G, 5VX6, 5KGO, 5EJL, 5XLY, 2L74, 5Y4R, 4RT0, 4RT1). In the PilZ domain of YcgR (PDBID: 5Y6F) the “DxSxxG” motif corresponds to D145, S147 and G150, and the “RxxxR” motif corresponds to R114 and R118, Figure 7D,E. The ligand was found as *closed dimers*, Figure 8. One PilZ was found to interact with a trimeric c-di-GMP (PDBID: 4XRN), Figure 8B.	[149,150]
	*E. coli* (5Y6F, 5Y6G)	YcgR like proteins such as the motility inhibitor (MotI) protein is a diguanylate receptor that binds c-di-GMP, acting as a molecular clutch on the flagellar stator MotA to inhibit swarming motility. The PilZ domain of MrkH, also a YcgR like protein, is transcriptional regulator protein, and binds c-di-GMP as well as DNA sequences to regulate type 3 fimbriae expression and biofilm formation. YcgR proteins regulate motility and biofilm formation by sensing c-di-GMP.		[151]
	*B. subtilis* (5VX6)			[152]
	*K. pneumoniae*. (5KGO, 5EJL)			[153,154]
	*P. aeruginosa*. (5XLY, 2L74, 5Y4R)	MapZ in complex with c-di-GMP interacts directly with a chemotaxis methyltransferase, CheR1, and inhibits its activity. In this manner, it regulates chemotaxis in *Pseudomonas aeruginosa*.		[54,155,156]
	*P. aeruginosa* (4RT0, 4RT1)	The alginate biosynthesis protein Alg44 regulates alginate secretion to promote biofilm formation by sensing dimeric c-di-GMP molecules.		[157]
	*P. aeruginosa* (4XRN)	Unknown function	The ligand is in an unusual trimeric oligomerization state, in which the six guanine bases are oriented almost parallel to each other, Figure 8B.	[158]
**I-site of GGDEF domains** (PF00990)	*P. fluorescens* (5EUH for GcbC)	Proteins containing GGDEF domains are DGCs and some of them are regulated by feedback regulation by interaction of c-di-GMP to their allosteric site (I-site).	Proteins with GGDEF domain act as receptor proteins when c-di-GMP binds their allosteric site via the RxxD motif. In the WspR GGDEF (PDB 3BRE) this motif corresponds to Arg242, Ser243, Ser244 and Asp245. The ligand was found as *closed dimers*, very similar to the PilZ proteins, Figure 8.	[159]
	*P. aeruginosa* (3BRE and 3I5C for WspR; 4EUV, 4ETZ, 4EU0 for PelD)			[160,161,162]
	*P. syringae* (3I5A for WspR)			[160]
	*M. hydrocarbonoclasticus* (3IGN for MqR89a)			[163]
	*T. maritima* (4URG, 4URS for TM1788)			[164]
	*C. vibrioides* (1W25, 2WB4, 2V0N for PelD)			[70,80,165]
	*E. coli* (3TVK, 4H54 for DgcZ)			[166]
	*P. aeruginosa* (4DN0)	PelD is a membrane protein in which the cytoplasmatic GGDEF domain binds c-di-GMP to regulate the synthesis of the PEL exopolysaccharide.		[167]
**Degenerate EAL domains** (PF00563)	*X. citri* (4FOK, 4FOJ, 4FOU)	The FimX protein regulates twitching motility by sensing c-di-GMP molecules through its EAL domain and regulates the type IV pilus machinery.	Proteins with EAL domain, such as FimX (PDB 4FOK), interact with the c-di-GMP by Q463, F479, L480, R481, S490, P491, M495, D508, R534, E653, F654, Q673, G674, D675 and T680. The A_478_F_479_L_480_ residues belong to a degenerate EAL motif, Figure 7H and I. The ligand was found always as *open or intermediate monomers*. Different EAL containing proteins bind the most diverse c-di-GMP conformation states analyzed in this review, Figure 8.	[168]
	*P. aeruginosa* (3HV8)			[169]
	*X. campestris* (4F3H, 4F48)			[170]
	*V. cholerae* (6PWK, 6IH1)	The transmembrane receptor LapD is a multidomain protein, in which the C-terminal EAL domain binds c-di-GMP to prevent cleavage of the surface adhesin LapA, inhibiting biofilm dispersal.		[171,172]
	*P. fluorescens* (3PJT, 3PJU)			[126]
**C-terminal domain of BldD** (PF not defined)	*S. venezuelae* (5TZD, RsiG protein: 6PFJ and, RsiG-σ^WhiG^ complex: 6PFV)	BldD is a master regulator of cell development. BldD represses the transcription of close to 170 sporulation genes during vegetative growth controlling morphological differentiation and also directly control expression of antibiotics. BldD has an N-terminus helix-turn-helix motif (HTH), while the C-terminal domain binds four c-di-GMP molecules to regulate cell differentiation.	The C-terminal domain of BldD (PDB 5TZD) interacts with a tetramer of c-di-GMP, forming a BldD_2_-(c-di-GMP)_4_ complex, by two motifs: R_114_G_115_D_116_ and R_125_Q_126_D_127_D_128_. The ligand was found as *closed tetramers*, Figure 8. A dimer of RsiG or RsiG in complex σ^WhiG^ binds (c-di-GMP)_2_ at the dimer interface and the ExxxSxxRxxxQxxxD motif of each helix of a coiled coil are involved in the ligand binding. The two repeats are: E_64_xxxS_68_xxR_71_xxxQ_75_xxxD_79_ and E_162_xxxS_166_xxR_169_xxxQ_173_xxxD_177_. The residues D106, S108, H110, S112 and R115 of RsiG also bind (c-di-GMP)_2_ as well as the K57, G61 and R62 of σ^WhiG^. The ligand was found as *intermediate dimer*.	[173,174]
	*S. coelicolor* (4OAZ)			[9]
**REC domain** (Response_reg, PF00072)	*V. cholerae* (3KLO)	VpsT is transcriptional regulator that binds c-di-GMP at its REC domain to control biofilm formation and motility. VpsT is described as a master regulator for biofilm formation and consists of an N-terminal REC domain and a C-terminal HTH domain.	A c-di-GMP_2_ binds into the VspD interface between two REC domains; the REC dimerization is required for ligand binding. Proteins with the REC domain of VpsT (PDB 3KLO) interact with two molecules of c-di-GMP by a K and a W[F/L/M][T/S]R motif that correspond to: K120, W131, L132, T133 and R134. The ligand was found as *closed dimers*, Figure 8.	[51]
**Pseudo-receiver Domain**	*C. vibrioides* (6QRL)	ShkA has a pseudoreceiver domain (Rec1) that binds c-di-GMP to allow the autophosphorylation and subsequent phosphotransfer and dephosphorylation of the protein. The c-di-GMP binds to the protein to release the C-terminal domain to step through the catalytic cycle.	C-di-GMP binds to the Rec1-Rec2 linker that contain the DDR motif. The residues involved in the ligand binding are: R324, Y338, I340, P342, R344, S347, Q351. The D369, D370 and R371 from the DDR motif located in a loop are inside of the c-di-GMP binding site in the apo form of the protein suggesting that c-di-GMP compete with this protein loop.	[175]
**T2SSE_N domain** (PF05157)	*V. cholerae* (5HTL)	MshE is an ATPases associated with the bacterial type II secretion system, homologous to the type IV pilus machinery. Its N-terminal domain binds c-di-GMP and cGAMP with different affinities, while the C-terminal catalytic domain binds ATP. The MshE N-terminal domain (T2SSE_N) binds c-di-GMP (*Kd* of 0.5 μM) with higher affinity than cGAMP (*Kd* of 330 μM).	The N-terminal domain of MshE (locus tag VC0405, PDB 5HTL) interacts with c-di-GMP by mainly two similar motifs spaced by five residues. These motifs have a similar sequence, RLGxx(L)(V/I)xxG(I/F)(L/V)xxxxLxxxLxxQ, and the residues involved to ligand binding are shown in bold and correspond to R_9_L_10_G_11_ and L_25_xxxL_29_xxQ_32_ for the motif I, and R_38_L_30_G_40_ and L_54_xxxL_58_xxQ_61_ for motif II. Other residues also important to ligand binding are: R7, D108 (from the C-terminal ATPase domain), and the main chain of D41. The ligand was found as *open monomers*, similar to those found in EAL domains, Figure 8.	[176]
**MerR domain** (PF00376)	*P. aeruginosa* (5XQL)	BrlR upregulates the expression of multidrug efflux pumps. c-di-GMP activates BrlR expression and enhances its affinity for binding DNA. BrlR has an N-terminus DNA-binding motif (HTH_MerR domain described in the Pfam as MerR domain), and a C-terminus effector-binding domain (GyrI-like domain) linked by a coiled-coil region.	There are two different c-di-GMP binding sites located at the N-terminus of the protein, mainly at the DNA binding domain of each BrlR protomer of the protein tetramer. Binding site 1 is composed of M1, R31, D35, Y40, and Y270. The binding site 2 is composed of P61, A64, R67, R70, F83, R86. The ligand was found as *closed monomers*, Figure 8.	[177]
**Sigma54_activat** (PF00158) or **AAA+_ATPase** (IPR003593)	*P. aeruginosa* (5EXX)	FleQ is a transcription regulator and a contains three domains: a central AAA+ ATPase σ(54)-interaction domain, flanked by a divergent N-terminal receiver domain and a C-terminal helix-turn-helix DNA-binding motif. FleQ binds c-di-GMP through itsAAA+ ATPase domain at a different binding site than the catalytic pocket site. FleQ regulates the expression of flagellar and exopolysaccharide biosynthesis genes in response to cellular levels of c-di-GMP.	FleQ binds c-di-GMP at the N-terminal part of the AAA+ ATPase through the L_142_F_143_R_144_S_145_ motif (R-switch), E_330_xxxR_334_ motif, and residues R185 and N186 of the post-Walker A motif KExxxRN. The ligand was found as *closed dimers*, Figure 8.	[57]
***HATPase_c*** (PF02518)	*C. vibrioides* (5IDM)	Cell cycle kinase CckA is a bifunctional histidine kinase/phosphatase enzyme, mediating both phosphorylation and dephosphorylation of downstream targets. CckA binds c-di-GMP and drives the cell cycle progression by swapping the CckA kinase activity into phosphatase mode.	CckA is a membrane and multidomain protein, in which a catalytically active (CA) domain binds c-di-GMP. The CA domain of cell cycle kinase CckA interacts with c-di-GMP by the residues Y514, K518, W523, I524, E550, H551, H552, H553, H554 and H555. The ligand was found as *open monomer*, Figure 8.	[178]
3′-5′ cGAMP or 3′-3′ cGAMP				
**STING** (TMEM173, PF15009)	*N. vectensis* (5CFM)	STING regulates the induction of type I interferons via recruitment of protein kinase TBK1 and transcription factor IRF3, activating IFN-β gene transcription. cGAS-STING responds to cytosolic DNA via binding to 3’-5’cGAMP.	STING proteins interact with cGAMP at the dimer interface. In the anemone STING (PDBID 5CFM), the residues involved with the ligand interaction are: Y206, R272, F276, R278, and T303 of each protomer of the dimer. Y280 binds the ligand by a water molecule. The ligand was found as *intermediate monomer*, Figure 8.	[144]
**c-di-GMP I Riboswitch** (RF01051)	*Geobacter* (4YAZ)	Acts as a transcriptional factor, switching between RNA secondary structures when bound to cGAMP, regulating its own expression. A human c-di-GMP I Riboswitch mutant (G20A) can also bind cGAMP.	3’-5’ cGAMP riboswitches bind cGAMP (PDBID 4YAZ) through the nucleotides G8, A11, A12, U13, A14, C15, A41, A42, G74, C75, and C76. The ligand was found as *closed monomer*, Figure 8.	[147]
	*Homo sapiens* (4YB1)			[147]
2’-3′ cGAMP				
**STING** (TMEM173, PF15009)	*Sus scrofa* (6A06)	STING regulates the induction of type I interferons via recruitment of protein kinase TBK1 and transcription factor IRF3, activating IFN-β gene transcription. The STING pathway plays an important role in the detection of viral and bacterial pathogens in animals.	STING proteins interact with 2’-3’ cGAMP produced by eukaryotic cGAS enzyme at the dimer interface. In the porcine STING (PDBID 6A06), the residues involved in ligand binding are: S162, Y167, I235, R232, R238, Y240, E260, and T263. The ligand was found as *closed monomer*, Figure 8.	[143]
	*Gallus gallus* (6NT7, 6NT8)			[179]
	*Rattus norvegicus* (5GRM)			[180]
	*N. vectensis* (5CFQ)			[144]
	*Homo sapiens* (4LOH, 4LOJ, 4KSY, 6DNK)			[136,181,182]
3′-5′ c-di-AMP				
**STING** (TMEM173, PF15009)	*Sus scrofa* (6A03, 6IYF)	STING binds eukaryotic 2’-3′ cGAMP with high affinity compared with bacterial CDNs such as c-di-GMP, c-di-AMP, and 3′-5′ cGAMP.	STING proteins interact with c-di-AMP in a different manner than c-di-GMP, but still at the same dimer interface. In the porcine STING (PDBID 6A03), the amino acids involved with the interaction are: S162, Y167, I235, R232, R238, Y240, and T263. The ligand was found as *closed monomers*, Figure 8.	[143]
	*N. vectensis* (5CFN)			[144]
	*H. sapiens* (6CFF and 6CY7)			[182]
	*Mus moluscus* (4YP1)			[183]
**Aldo-keto reductase** (PF00248)	*Mus musculus* (5UXF)	RECON (reductase controlling NF-κB) is an aldo-keto reductase and a STING antagonist. It negatively regulates the NF-κB activation that induces the expression of IFN-induced genes. RECON recognizes c-di-AMP by the same site that binds the co-substrate nicotinamide. One AMP molecule (AMP1) of c-di-AMP has essentially the same position as the AMP portion of the NAD+ co-substrate, while another AMP (AMP2) presents a shifted position.	RECON binds c-di-AMP by the residues: E276, E279, N280, L219, and A253 in contact with AMP1, while Y24, Y216, Y55, and L306 are in contact with AMP2. L219, T221, and G217 are also involved in ligand binding. The ligand was found as open *monomers*, Figure 8.	[62]
**c-di-GMP I Riboswitch** (RF01051)	*E. coli* (G20A/C92U mutant Riboswitch, 3MUV)	Bacterial c-di-AMP is involved in cell wall stress and signaling DNA damage through interactions with several protein receptors and a widespread *ydaO*-type riboswitch, one of the most common riboswitches in various bacterial species. This riboswitch is found in the vicinity of genes involved in cell wall metabolism, synthesis and transport of osmoprotectants, sporulation and other important biological processes [127,128]. A c-di-GMP I Riboswitch mutant (G20A/C92U, PDB 3MUV) can also bind c-di-AMP.	*ydaO* riboswitch (PDBID 3MUV) binds c-di-AMP molecules into two binding sites: site 1 (G5, C6, C7, G8, A45, G68, G69, A70, U71, A72, C82, C83, G107, C108, and A109) and site 2 (A9, G23, G24, A25, G26, G41, G42, U43, C88, C89, A93, G102, AND G103). The ligand was found as *closed monomers*, Figure 8.	[145]
**ydaO-yuaA Riboswitch** (RF00379)	*T. pseudethanolicus* (4QK8 and 4QKA)			[184]
	*T. lienii* (4QK9)			[184]
	*B. subtilis* (4W92 and 4W90)			[185]
	*C. subterraneus* (4QLM and 4QLN)			[186]
	*H. sapiens* (6N5K, 6N5L, 6N5N, 6N5O, 6N5P, 6N5Q, 6N5R, 6N5S and 6N5T)			[187]
**Cyclic-di-AMP receptor** (PF06153)	*S. aureus* (4WK1 and 4D3H)	PII-like signal transduction protein (PtsA) is a c-di-AMP receptor. PII-like proteins are associated with nitrogen metabolism using different pathways. PtsA binds c-di-AMP with a *Kd* of 0.37 µM (intracellular c-di-AMP is in μM range). Others c-di-AMP receptors bind the ligand with a *Kd* range of 0.1 to 8 μM.	PstA (PDBID 4D3H) forms trimers and binds to c-di-AMP at the interface between two molecules through interactions with the residues N24, R26, T28, A27, F36, L37, N41, G47, F99, and Q108. The ligand was found as *intermediate monomer*, Figure 8.	[188,189]
	*L. monocytogenes* (4RWW)			[61]
	*B. subtilis* (4RLE)			[190]
**Pyruvate carboxylase** (HMGL-like, PF00682)	*L. lactis* (5VYZ and 5VZ0)	*L. monocytogenes* pyruvate carboxylase (LmPC) or *L. lactis* pyruvate carboxylase (LIPC) are inhibited by c-di-AMP. LmPC is biotin-dependent enzyme with biotin carboxylase (BC) and carboxyltransferase (CT) activities. c-di-AMP causes conformational changes in the CT dimer that may explain the molecular mechanism for its inhibitory activity.	LIPC forms a tetramer and each c-di-AMP molecule binds at a protein dimer interface at the carboxyltransferase (CT) domain (HMGL-like domain in the Pfam) (PDBID 5VYZ) in a binding site that is not well conserved among pyruvate carboxylases. The residues involved in the interaction are: Q712, Y715, I742, S745, G746, and Q749 from both monomers. The ligand was found as *intermediate monomers*, Figure 8.	[135]
	*L. monocytogenes* (4QSH and 4QSK)			[33]
**TrkA**_C (PF02080)	*S. aureus* (4YS2, 4XTT, and 5F29)	Potassium transporter A (KtrA) and Bacterial cation-proton antiporter (CpaA) are members of the RCK domain family of proteins (Regulator of conductance of K^+^) and regulates the cellular potassium conductance. The C-terminal domain (RCK_C or TrkA_C) binds specifically c-di-AMP molecules (*Kd* of 43.1 nM), causing inactivation of the KtrA.	c-di-AMP binds at the RCK_C domain of KtrA in the interface of a dimer (PDBID 4XTT). The residues involved in the interaction are I163, I164, D167, I168, R169, A170, N175, I176, and P191 from both monomers. R169 and the isoleucine residues (hydrophobic pocket) are well conserved in other species. The ligand was found as *closed monomers*, Figure 8.	[59,183,191]
**CBS domain** (PF00571)	*L. monocytogenes* (5KS7)	Intracellular pathogen *L. monocytogenes* synthesizes and secretes c-di-AMP during growth in culture and also in host cells. Overexpression of c-di-AMP is toxic to the cell. c-di-AMP binds to OpuC carnitine transporter at the CBS domain (*Kd* of 4.8 μM), probably inhibiting carnitine uptake. OpuC is the ATPase subunit of the transporter complex OpuCA.	c-di-AMP binds to the cystathionine β-synthase domain (CBS) of OpuC at the dimer interface. The residues involved in ligand binding are well conserved among OpuCA orthologues and are composed by the following residues: V260, V280, T282, Y342, I355, I357, R358, and A359. The ligand was found as *open monomers*, Figure 8.	[24]

### Conformation of Cyclic Dinucleotides inside the Binding Site of Receptors

The cyclisation between two nucleotides of the most common CDNs involves the formation of a phosphodiester bond that links the C3’ of one pentose ring with the C5’ of another, resulting in a 3’-5’ cyclic dinucleotide. This kind of cyclisation creates a two-fold symmetry between two pentose rings of dinucleotides. Only cGAMP has been reported to present not only a 3’-5’ linkage, but also being found with a 2’-3’ one that contains two distinct phosphodiester linkages, one between C3′ of AMP and C5′-phosphate of GMP, and the other between C5′-phosphate of AMP and C2’ of GMP (Figure 8A).

The dinucleotides can assume different conformations in the binding site of different receptors that can be described in relation to the base and the ribose conformations. The ribose ring can assume three different configurations, C3’-endo, C2’-endo, or C2’-exo. When taking into account receptor structures in complex with cyclic dinucleotides (Table 1), more than 80% of the ligands have the two pentose rings in C3’-endo, almost 15% have one of the pentoses in C3’-endo and the other in C2’-endo and only one structure has the two pentoses in C2’-exo configuration (Figure 8A). Furthermore, the base can assume a *syn* or *anti* conformation in relation to the pentose by the N-glycosidic bond, and only one of the structures, the FimX EAL domain from *Xanthomonas citri* (PDBID: 4FOK) [168] has one of the base at the *syn* conformation, which is the less stable state of the molecule. The conformation C3’-endo/C3’-endo is the more representative for the c-di-GMP and c-di-AMP molecules, while cGAMP is preferentially in C3’-endo/C2’-endo conformation (Figure 8A).

The overall conformation of the ligand can be classified in three conformations with respect to the base proximity: 1—*closed* conformation (shaped as a horseshoe) is when the two base rings are face-to-face; 2—*open* conformation is when the two base rings are far from each other in an elongated conformation and; 3—*intermediate* conformation (shaped as a boat) is when the bases are not in the *closed* conformation or in the *open* conformation (Figure 8A). The comparison of c-di-AMP conformations in the c-di-AMP receptors binding sites was described by Chin and collaborators and they conclude that c-di-AMP molecules are bound in two main conformational types, "U-shape" or "V-shape" that correspond to *closed* and *intermediate* conformation, respectively [183]. The comparison of c-di-GMP conformations in the biding sites of c-di-GMP receptors was described in detail by Chou and Galperin [193] and by Schirmer [67]. In both papers, c-di-GMP molecules are found in the protein binding sites in different conformational types ranging from fully stacked form (*closed conformation*) to an extended form (*open conformation*) allowing significant binding flexibility. The c-di-GMP bases may interact with the protein binding site by stacking with arginine or phenylalanine/tyrosine residues through the hydrophobic surface of the base. The c-di-GMP bases may also interact with acidic residues (aspartate or glutamate) through Watson–Crick–edge interaction or with arginine residue through Hoogsteen–edge interaction [193]. 

The c-di-GMP molecule in solution is found in a fast equilibrium between a monomeric state and as a dimer with intercalated bases with a *Kd* of about 1 mM under physiological salt conditions [194]. Nevertheless, the intracellular concentration of c-di-GMP is about the μM range, suggesting that free c-di-GMP molecules are monomeric inside of the cells and c-di-GMP dimers, even though being found in some c-di-GMP receptor pockets, are probably not relevant for c-di-GMP signaling [67]. Looking at the conformation of c-di-GMP when it is bound to proteins, which include its receptors and the active site of DGCs enzymes that contain GGDEF domains, most of them are found as monomers or dimers, though trimeric and tetrameric structures were also observed in PilZ [158] and the C-terminal domain of BldD proteins [173], respectively (Figure 8B). Interestingly, PilZ is the only one that binds c-di-GMP in monomer, dimer, and trimer forms, while EAL domain binds c-di-GMP monomers with the largest conformational divergences. Proteins containing STING domain and RNA riboswitches are bound to CDN monomers that share similar conformations (Figure 8B,D). Looking at the conformation of c-di-AMP and cGAMP when bound to proteins or riboswitches, all of them are found as monomers (Figure 8C,D). Therefore, even though bacteria have a large class of specific CDN receptors, which include not only proteins but also RNAs; surprisingly, the conformations of the ligands at the binding site are similar.

## 4. Distribution of Proteins Containing GGDEF and DAC Domains in Bacteria

Initial reviews of the distribution of DisA homologs across bacterial clades suggested that c-di-AMP would play a more important role in Gram-positive bacteria than in Gram-negative and that, in general, bacteria would avoid allowing these two signaling networks to co-exist, so as to avoid unintended crosstalk and to easily regulate the balance of these second messengers within the cell [21,22]. Subsequent surveys on the distribution of DAC and GGDEF homologs don’t support the idea that DAC homologs are rare among Gram-negative bacteria, as members of lineages such as Cyanobacteria, Spirochaetes, and Deltaproteobacteria often carry both DAC and GGDEF genes, a profile compatible with the complex lifestyles and genomes of these lineages. In addition, among Gram-positives, most members of Firmicutes and Actinobacteria, including model organisms such as *Bacillus*, *Clostridium*, *Streptomyces*, *Listeria*, and *Mycobacterium*, produce both signaling molecules and possess a wide array of GGDEF genes, following the general trend of having close to as many genomes with both DAC and GGDEF as possible (see Appendix A and Figure 9). The only lineages were several of the genomes sampled that seem to have at least one DAC homolog, but no or very few and rare recognizable GGDEF homologs are Bacteroidetes and the Archaea. In both lineages, the number of genomes with both DAC and GGDEF falls below 50% of the maximum allowed, i.e., the smallest between the number of genomes carrying DAC or GGDEF. Genomic data strongly suggest that there is a tendency for bacterial cells to use both c-di-AMP and c-di-GMP signaling networks simultaneously, which would imply that both the control of their synthesis and turnover and the specificity of their sensors are carefully tuned.

## 5. Conclusions 

Recently, our knowledge about cyclic dinucleotide second messengers has been under expansion with the discoveries of new CDNs. At the moment, three different classes of prokaryotic proteins are known to synthesize CDN molecules: (i) proteins containing GGDEF domain that synthesizes mainly c-di-GMP; (ii) CD-NTases enzymes that have the catalytic domain known as SMODS and synthesizes mainly cGAMP; and (iii) DAC proteins that have a catalytic domain called DAC domain (also described as DisA_N domain) that synthesizes mainly c-di-AMP. These CDN synthetases do not share structural similarities, use different residues for substrate binding, and probably possess different catalytic mechanisms suggesting that they probably evolved independently to catalyze similar chemical reactions.

As evidence of the importance and ubiquity of bacterial CDNs, it is interesting to note that mammalian cells evolved to sense bacteria by detecting these molecules to stimulate the immune system to counterattack infections. Thus, the use of CDNs as adjuvants in vaccines has been considered, as they can be used as stimulators of the innate immune system [63,195].

The huge repertoire of CDN receptors and the complexity of CDN signaling networks in bacteria are shown in this review. Surprisingly, different CDNs share conformational similarities even in the pocket of different classes of receptors. CDNs can regulate bacterial behaviors at different speed levels, directly regulating protein function for a faster response or, more slowly, by affecting gene transcription or protein translation. Remarkably, different CDN second messenger systems may coexist in many organisms, which would imply that both the control of their synthesis and turnover and the specificity of their sensors are carefully tuned.

Therefore, the new discoveries reviewed in this paper open up questions about how bacteria coordinate the three mains bacterial CDNs: are they interconnected to regulate the same bacterial phenotype, or do they act independently? Do bacteria use the three CDNs as second messengers or is one chosen? Are the CDN signaling pathways conserved in different bacteria? Will other CDNs be discovered to also be second messengers? 

## Figures and Tables

**Figure 3 molecules-25-02462-f003:**
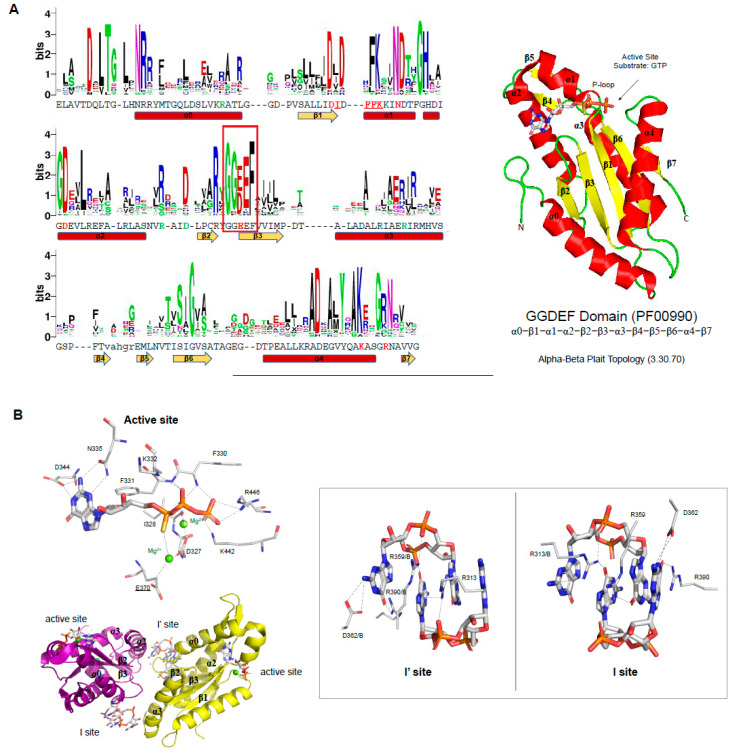
Conserved sequences within GGDEF members and their active and I-sites pockets. (**A**) residue frequency in GGDEF domains. Using the Dali server [79], 23 sequences of GGDEF domain structures were used to create a multiple sequence alignment, and the sequence logo was created with the WebLogo server [90]. The sequence shown below the logo and the secondary structure elements belong to PleD of *Caulobacter vibrioides* (PDBID: 2V0N). Residues colored in red are involved in ligand or magnesium binding (for underlined residues, only the main chain is involved) and those colored in green are located in the I-sites. The GGDEF motif is placed in a red box. On the right, the structure of the GGDEF domain of PleD is shown as a cartoon. The topology of GGDEF is shown below the structure, and the CATH topology name and code are also shown [91]; (**B**) interaction network between the GGDEF domain of PleD binding pocket with the substrate, GTP. In the bottom, the PleD structure in the inactive conformation is shown, in which the two inhibitory sites are shown (I-site and I’-site). On the right, it is shown in more detail the residues involved in the (c-di-GMP)_2_ interactions at the inhibitory sites. Gray dotted lines represent hydrogen bonds. The magnesium ions are colored in green. GTP and the protein residues involved in its binding are shown as sticks. Carbons are colored white, oxygens are red, nitrogen atoms are blue, and phosphorous atoms are orange.

**Figure 4 molecules-25-02462-f004:**
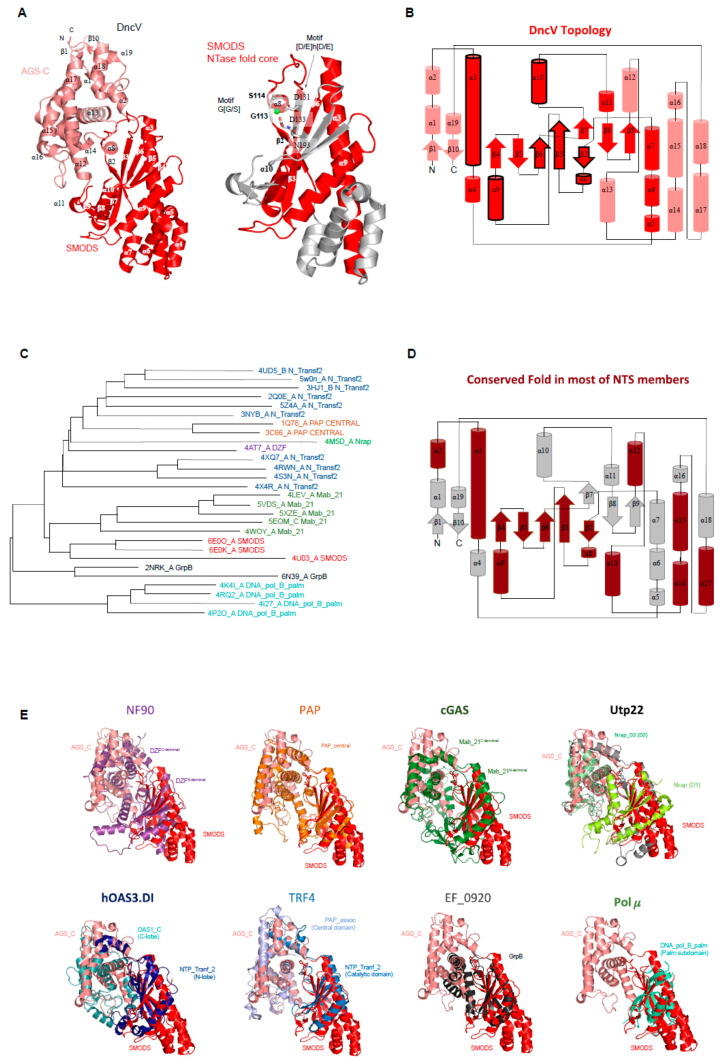
Structural similarities between the SMODS domain and other nucleotidyltransferase superfamily members. (**A**) the N-terminal domain SMODS (colored in red) and the C-terminal ACS_C (light pink) of the DncV protein structure are shown as cartoons (PDBID: 4U0M). The SMODS domain is involved in cGAMP synthesis and belongs to the nucleotidyltransferase superfamily (NTS). The NTS fold is characterized by the presence of a minimal conserved core of a mixed β-sheet flanked by α-helices with α1-β1-α2-β2-α3-β3-α4 topology that correspond to α3-β2-α8-β3-α9-β6 (colored in red), missing the α4 element. Various insertions are observed and are colored in grey (right panel). Members of NTS contain three conserved motifs located at the active site: (i) G[G/S] located at α8, (ii) [D/E]h[D/E] (h indicates a hydrophobic amino acid) located at β3, and (iii) [D/E] located at β6. Two of these motifs are conserved in proteins containing SMODS domains (see Figure 5A). (**B**) topology of DncV, showing the location of the NTS fold core (bold outlines). The secondary structure elements from SMODS are colored in red and the AGS-C domain in light pink. (**C**) dendrogram showing structures similar to DncV made with the Dali server [79] (query: DncV, PDBID: 4U03). Each domain is colored with different colors and the PDBID_chain and the Pfam name are shown for each branch. (**D**) The conserved fold found in most of these structures, all NTS members, is shown in brown overlaid on the DncV topology. (**E**) structural superposition of DncV (PDBID: 4U03) with other NTS members found in the panel (**C**) dendrogram, using the same colors to represent each domain. At the top of each structural alignment is shown the protein’s name. NF90 is colored in purple (PDBID: 4AT7), PAP is colored in orange (PDBID: 1Q78), cGAS is colored in dark green (PDBID: 5XZE), Utp22 is colored in light green (PDBID: 4M5D), hOAS3.DI is colored in blue (PDBID: 4S3N), TRF4 is colored in light blue (PDBID: 3NYB), EF_0920 is colored black (PDBID: 2NRK), and Pol µ is colored in green (PDBID: 2IHM).

**Figure 6 molecules-25-02462-f006:**
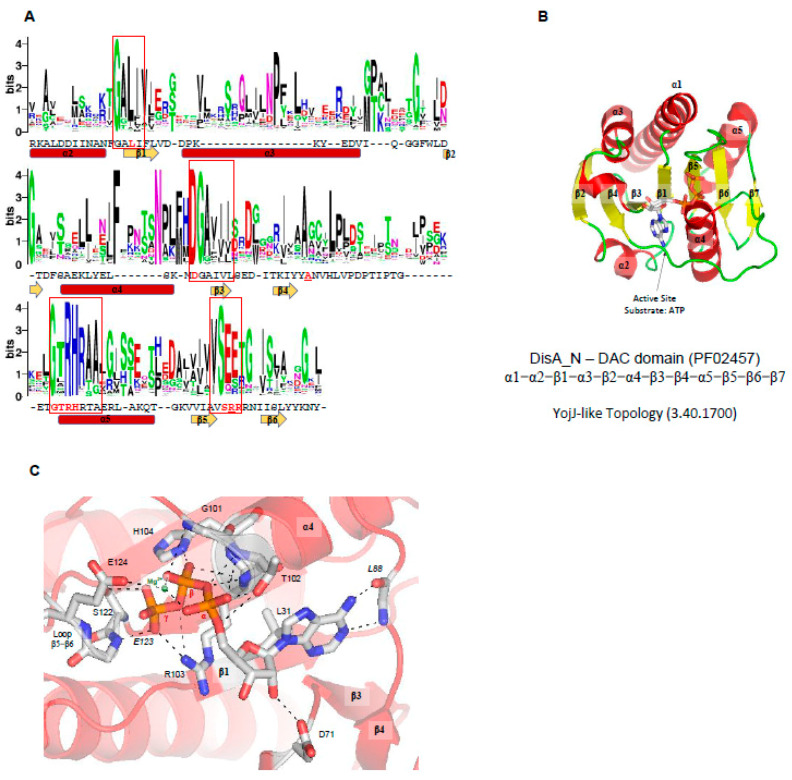
Conserved sequences within DAC domain members and its substrate binding pocket. (**A**) residue frequency present in DAC proteins. In addition, 609 sequences were used from the Pfam database to create a multiple sequence alignment of the DAC domain of different proteins using ClustalW [116], and the sequence logo was done using the WebLogo server [90]. The sequence shown below the logo and the secondary structure elements belong to the DisA protein from *T. maritima* (PDBID: 3C1Y). Residues that bind ATP or the magnesium cation are colored in red, underlined residues bind mainly by the main chain. Conserved motifs within DAC members are placed in red boxes: GALI, DGAhh, GxRHRxA, and (V/I)SEE motifs. (**B**) structure of the DAC domain of CdaA from *Listeria monocytogenes* (PDBID: 4RV7). The substrate ATP is found bound at the active site. The DAC domain topology is shown below the structure, and the CATH topology name and code are also shown [91]; (**C**) interaction network between the CdaA binding pocket with the substrate, ATP (PDBID: 4RV7). Gray dotted lines represent hydrogen bonds, the magnesium ion is colored in green, and the ATP and the protein residues involved in its binding are shown as sticks. Carbons are colored white, oxygens are red, nitrogen atoms are blue, and phosphorous atoms are orange.

**Figure 7 molecules-25-02462-f007:**
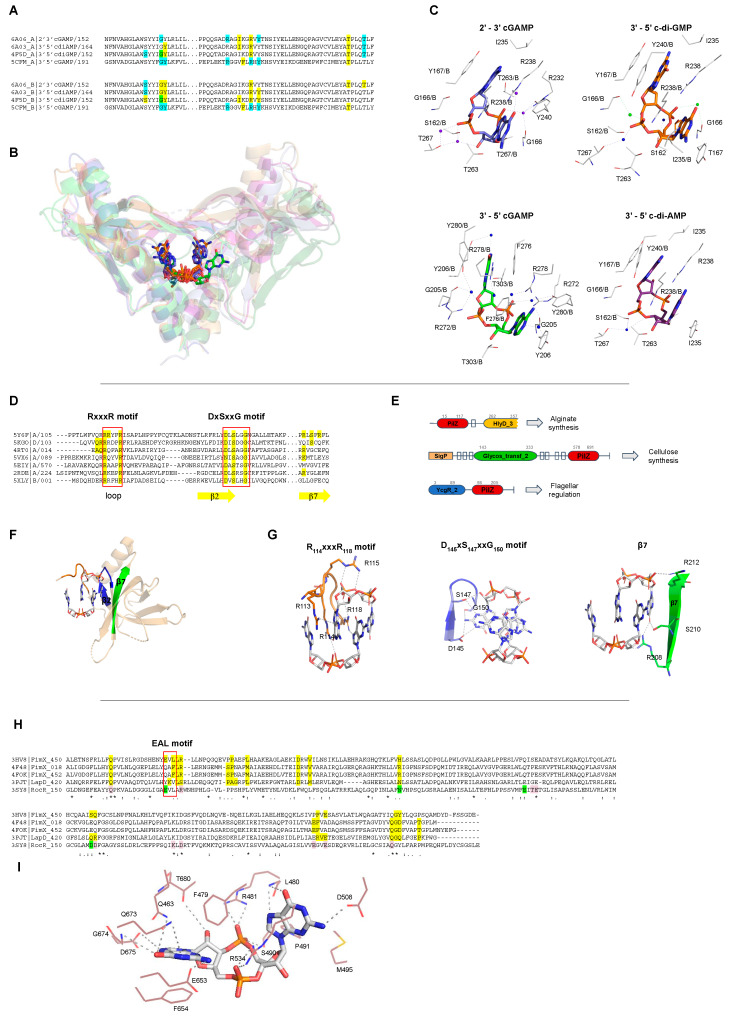
Some CDN receptors: STING, PILZ, and degenerate EAL domains. (**A**) multiple sequence alignment of STING proteins in complex with different CDNs: monomeric 3’-5’ c-di-GMP (PDBID: 4F5D), 2’-3’ cGAMP (PDBID: 6A06), 3’-5’ c-di-AMP (PDBID: 6A03), and 3’-5’ cGAMP (PDBID: 5CFM). The residues highlighted in yellow are involved with direct interaction with the ligand, those in green are involved with interaction with the ligand via magnesium ions, and those in cyan interact via water molecules; (**B**) structural superposition of STING proteins (domain TMEM173) in complex with different CDNs bound at the same protein interface. The ligands are colored by element: nitrogen atoms are in dark blue and oxygens are in red. Carbons are colored according to the ligand: 3’-5’ c-di-GMP in orange; 2’-3’ cGAMP in blue, 3’-5’ c-di-AMP in purple and; 3’-5’ cGAMP in green; (**C**) residues involved in interactions with the different ligands are the same one highlighted in panel (**A**). The residues are colored by element, with carbon in white, nitrogen in dark blue, and oxygen in red, while the ligands are colored as described in panel (**B**). Water molecules are shown as blue spheres, while magnesium cations are shown as green spheres. (**D**) multiple sequence alignment of the PilZ domains that had their structures solved in complex with c-di-GMP molecules. Residues highlighted in yellow are involved in interactions with the ligand. The secondary structure elements shown belong to the PilZ domain of YcgR from *E. coli* (PDBID: 5Y6F). The motifs conserved within PilZ members are placed in red boxes; (**E**) some domain organizations found in proteins containing PilZ domains (Alg44, BcsA, and YcgR) and their related functions; (**F**) structural representation of a PilZ domain as a cartoon (PilZ domain of YcgR, PDBID: 5Y6F). Residues belonging to the “RxxxR” motif are colored in orange, the “DxSxxG” motif is colored in blue, and the β strand 7 is colored in green. The dimeric c-di-GMP is shown as sticks. The interaction network presented in this figure is shown in more details in panel (**G**); (**H**) multiple sequence alignment of proteins containing degenerate EAL domains (PDBID: 4F48, 4FOK, and 3PJT) and a catalytic EAL domain (PDBID: 3SY8) that had their structure solved in complex with monomeric c-di-GMP. The residues highlighted in yellow are involved with direct interactions with the ligand, those in green are involved with interactions with the ligand via magnesium ions. In the case of RocR, which has a catalytic EAL domain, the residues highlighted in salmon were experimentally demonstrated to be important for catalysis [123].The consensus “EAL” motif is placed in a red box; (**I**) interaction network of the binding site of a FimX degenerate EAL domain (PDBID: 4FOK). Gray dotted lines represent hydrogen bonds. The residues and the c-di-GMP molecule are colored by element. The multiple sequence alignments were performed using the CLUSTAL W server [116].

**Figure 8 molecules-25-02462-f008:**
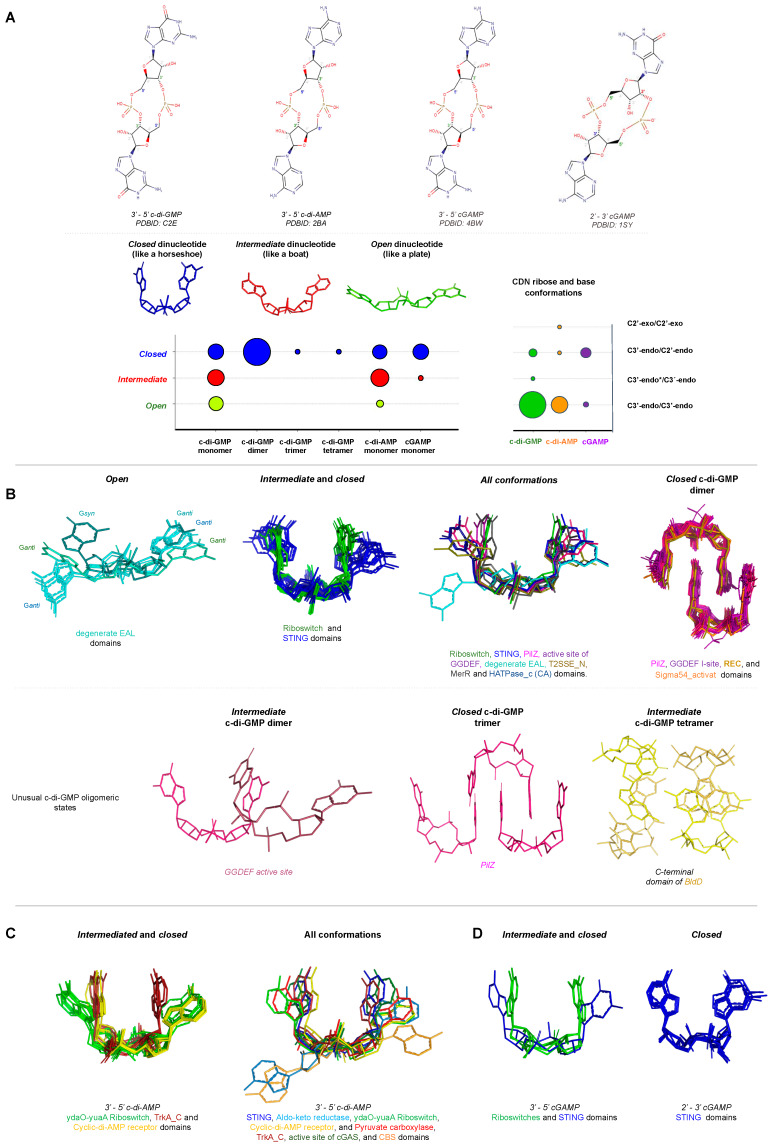
Diversity of cyclic dinucleotides produced by different organisms. All structures were observed within the three-dimensional protein structures deposited in the Protein Data Bank. (**A**) top panel: two-dimensional representation of different cyclic dinucleotides produced mainly by bacteria, with the exception of the 2’-3’ cGAMP molecule that is produced by eukaryotic cells by cGAS enzymes. The linkages between the pentoses and the phosphates are shown in green, blue, or red, and those carbons colored in grey are not involved in the phosphate linkage cyclisation. The structures of each CDNs were initially downloaded in the SDF format at the CHEBI website [192] and edited using MarvinSketch version 19.18. The PDBID of each CDN is also described for each ligand. Bottom panel: The three main conformations of CDNs found in the protein binding pockets in respect to the base proximity are described as: 1-*closed* conformation (shaped as a horseshoe), when the two base rings are face-to-face, colored in blue; 2-*intermediate* conformation (shaped as a boat), when the bases are not in the *closed* conformation neither in the *open* conformation, colored in red; and 3-*open* conformation (shaped as a plate), when the two base rings are far from each other in an elongated conformation, colored in lemon green. At the bottom of these structures is shown a bubble chart showing the frequency of each CDN in the *Closed*, *Intermediate*, and *Open* conformations. c-di-GMP is the only one that has been found in protein structures in different oligomeric states: as a monomer, dimers, trimers, and tetramers. The bubble chart on the right shows the ribose conformations that can be found in 3 different configurations, C3’-endo, C2’-endo, or C2’-exo, and the frequency of each of these conformation in the CDNs found in protein binding pockets. * The G*syn* conformation of the base ring in relation to the pentose is found only in one c-di-GMP structure (PDBID: 4FOK). Panels (**B**–**D**) show superpositions of different CDNs showing the heterogeneity of conformations found in the protein and riboswitch binding pockets; (**B**) top panel: different conformations for c-di-GMP found in different protein binding pockets and riboswitches shown as a superposition between them. 3’-5’ c-di-GMP structures found in degenerate EAL domains are colored in cyan (PDBID: 3HV8, 4F3H, 4F48, 3PJT, 3PJU), in riboswitches are colored in green (PDBID: 3Q3Z, 3MXH, 3MUT, 3MUR, 3MUM, 3IRW, 4YB0, 3IWN), in STING proteins are colored in blue (PDBID: 4EF4, 4EMT, 6RM0, 6S86, 4F9G, 4F5D, 4F5Y, 6A04, 5CFL, 5CFP), in PilZ domains are colored in pink (dimeric c-di-GMP: PDBID: 4ZMN, 5EUH, 3BRE, 3I5C, 1W25, 2WB4, 2V0N, 3TVK, 3I5A, 3IGM, 4URG, 4URS. Trimeric c-di-GMP, PDBID: 4XRN), in the active site of GGDEF domains (monomeric c-di-GMP, PDBID: 4RT1), and in the GGDEF I-site (dimeric c-di-GMP, PDBID: 5EIY, 5EJ1, 5EJZ, 4P00, 4P02, 5KGO, 5EJL, 5VX6, 5Y4R, 5XLY, 2L74, 4RT0, 5Y6F, 5Y6G) are colored in purple, in the T2SSE_N domain is colored in brown (PDBID: 5HTL), in the HATPase_c (CA) domain is colored in blue (PDBID: 5IDM), in the REC domain is colored in yellow (PDBID: 3KLO), and in the Sigma54_activat domain is colored in orange (PDBID: 5EXX). The unusual c-di-GMP oligomeric states found in one GGDEF active site is colored in pink and brown (PDBID: 3QYY), and in the C-terminal domain of BldD is colored in yellow (PDBID: 4OAZ); (**C**) different conformations for 3’-5’ c-di-AMP found in different protein binding pockets and riboswitches shown as a superposition between them. 3’-5’ c-di-AMP structures found in riboswitches are colored in green (PDBID: 4QK8, 4QK9, 4W92, 4W90, 4QLM, 4QLN, 4QKA), in TrkA_C domains are colored in brown (PDBID: 4YS2, 4YP1, 5F29), in Cyclic-di-AMP receptor domains are colored in yellow (PDBID: 4WK1, 4D3H, 4RWW, 4RLE), in a STING protein is colored in dark blue (PDBID: 6IYF), in an Aldo-keto reductase domain is colored in light blue (PDBID: 5UXF), in a Pyruvate carboxylase domain is colored in red (PDBID: 5VZ0), in the active site of cGAS is colored in dark green (PDBID: 3C1Y), and in a CBS domain is colored in orange (PDBID: 5KS7); (**D**) 3’-5’ cGAMP is found in *intermediate* and *closed* conformations in the ligand binding pocket of riboswitches, colored in green (PDBID: 4YAZ and 4YB1), and in a STING protein, colored in blue (PDBID: 5CFM). 2’-3’ cGAMP is found in a *closed* conformation in the ligand binding pocket of STING proteins, colored in blue (PDBID: 6NT7, 6NT8, 5CFQ, 4LOH, 4LOJ, 5GRM, 4KSY, and 6A06).

**Figure 9 molecules-25-02462-f009:**
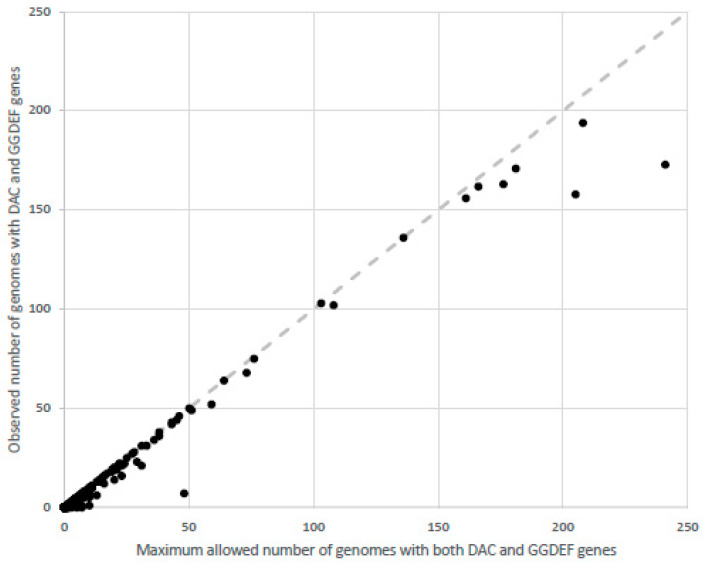
Lack of anti-correlation in the distribution of DAC and GGDEF genes per prokaryotic clades. Each dot represents a prokaryotic class, such as Gammaproteobacteria or Bacilli, as defined in the NCBI’s Taxonomy Database. For each class, the number of genomes harboring at least one DAC and one GGDEF gene and the number of genomes harboring both was calculated. If, for a given class, we consider the number of genomes with DACs and the number of genomes with GGDEF, the smallest of these numbers is the maximum number of genomes that could, in principle, carry both genes. That number is seen on the horizontal axis while the actual number of genomes carrying both genes is on the *y*-axis. These numbers are very close to the diagonal line, indicating that, in most cases, if members of a given lineage are carrying both DAC and GGDEF, they tend to keep both genes, instead of having to choose between them.

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
