# Peer review of "The World of Cyclic Dinucleotides in Bacterial Behavior"

_molecules, 2020, doi:10.3390/molecules25102462_

Round 1

Reviewer 1 Report

The manuscript by Purificação and colleagues presents a useful review of the structures of cyclic dinucleotide-related proteins: synthases, hydrolases and cyclic dinucleotide receptors. The manuscript is well written, the material is presented in a clear and logical way. I have only several minor comments.

1. The manuscript is full of minor errors in English, mostly in the usage of verbs in singular and plural form, use of both the (correct) “dinucleotide” and the (incorrect) “di-nucleotide” versions, and similar minor language problems. This text would need a careful copy editing by a native speaker of English.

2. The diversity of c-di-AMP and c-di-GMP conformations has been previously discussed in detail by, respectively, Chin et al (2015), PMID: 26171638, and Chou and Galperin (2016), PMID: 26055114. Omission of these references gives the impression that the current manuscript is the first to discuss these conformations. Ref. 187 is incomplete.

3. Table 1 is very useful but needs to be checked for completeness. As an example, it misses the structure of human STING–c-di-AMP complex, PDBID: 6cff, described by Ergun et al. (2019), PMID: 31230712.

Author Response

Reviewer:  The manuscript is full of minor errors in English, mostly in the usage of verbs in singular and plural form, use of both the (correct) “dinucleotide” and the (incorrect) “di-nucleotide” versions, and similar minor language problems. This text would need a careful copy editing by a native speaker of English.

Response: The manuscript was carefully revised and several errors in verbal agreement were corrected. The di-nucleotides were corrected for dinucleotide as suggested by the reviewer.

Reviewer:  The diversity of c-di-AMP and c-di-GMP conformations has been previously discussed in detail by, respectively, Chin et al (2015), PMID: 26171638, and Chou and Galperin (2016), PMID: 26055114. Omission of these references gives the impression that the current manuscript is the first to discuss these conformations. Ref. 187 is incomplete.

Response: The references were corrected. thanks for the correction and the following information has been added to line 959: "The comparison of c-di-AMP conformations in the c-di-AMP receptors binding sites was described by Chin and collaborators and they conclude that c-di-AMP molecules are bound in two main conformational types, "U-shape" or "V-shape" that correspond to closed and intermediate conformation, respectively[183]. The comparison of c-di-GMP conformations in the biding sites of c-di-GMP receptors was described in detail by Chou and Galperin [193] and by Schirmer [83]. In both papers c-di-GMP molecules are found in the protein binding sites in different conformational types ranging from fully stacked form (closed conformation) to an extended form (open conformation) allowing significant binding flexibility. The c-di-GMP bases may interact with the protein binding site by stacking with arginine or phenylalanine/tyrosine residues through the hydrophobic surface of the base. The c-di-GMP bases may also interact with acidic residues (aspartate or glutamate) through Watson-Crick-edge interaction or with arginine residue through Hoogsteen-edge interaction [193]. "        

Reviewer:  Table 1 is very useful but needs to be checked for completeness. As an example, it misses the structure of human STING–c-di-AMP complex, PDBID: 6cff, described by Ergun et al. (2019), PMID: 31230712.

Response: The table 1 was revised and some extra structures were added.

Reviewer 2 Report

The review presented by Dias et al. summarizes structural biology data of CDNs that enable signaling in bacteria.

This manuscript does not fulfill the requirements for a review in „Molecules“ although the topic is interesting.

In general, I strongly recommend and encourage the authors to reorganize and simplify the details of this review to reach a broader audience for this interesting topic. In that context, I suggest to end each chapter with a comprehensive conclusion that leads to the next chapter. This is completely missing, instead the authors present a lot of data without an interpretation of the data. The manuscript would be significantly improved in doing so. The manuscript is difficult to read and confusingly written. Therefore, my suggestion ist o rewrite it completely.

Even the abstract is quite confusing so that it remains an open question what the real message of this review might be. Moreover, I suggest to extract the necessary data to present the novel discoveries.

There are serious mistakes  and flaws in this manuscript e.g. a second messenger seems to be interchanged with receptors.The manuscript has to be rewritten completely and the English to be improved. There are lots of typos and grammatical errors. The sentences are too long which makes it difficult to follow up. The titles of the legends are not clear and the legends themselves are too long.

Parts of the conclusions have to be replaced into the introduction. Conclusively, I cannot recommend this manuscript for publication.

Minor Comments:

Line 14: “this world“ …this is not a scientific expression

Line 39: it should be Streptomycetes

Line 53: dinuleotides in special roles

Line 133: c-di-GMP are not show

109: cannot be describe

123: and deposited in protein data bank

301: The SMODS Domain

220-224: please make two or three sentences out of it. It is hard to understand what the authors reaaly mean.

694-695: …due to it being the first CDN as a bacterial messenger.. What does that mean?

Author Response

Dear Reviewer, Thanks a lot for your careful reading of the manuscript and for the constructive suggestions. Following are the answers for your questions or suggestions.

Reviewer: The review presented by Dias et al. summarizes structural biology data of CDNs that enable signaling in bacteria. This manuscript does not fulfill the requirements for a review in „Molecules“ although the topic is interesting.

Response: I agree with the reviewer that the topic of dinucleotides cyclic is not a commonly topic described by the "Molecules" journal. Nevertheless, in my opinion, the topic is in the scope of journal to inspire scientific community in the role of CDNs controlling bacterial behaviors.

Reviewer: In general, I strongly recommend and encourage the authors to reorganize and simplify the details of this review to reach a broader audience for this interesting topic. In that context, I suggest to end each chapter with a comprehensive conclusion that leads to the next chapter. This is completely missing, instead the authors present a lot of data without an interpretation of the data. The manuscript would be significantly improved in doing so. The manuscript is difficult to read and confusingly written. Therefore, my suggestion ist o rewrite it completely.

Response: We include a comprehensive conclusion at the end of each chapter and a linker to the next chapter. The manuscript was carefully revised and several errors in verbal agreement were corrected.

Reviewer:  Even the abstract is quite confusing so that it remains an open question what the real message of this review might be. Moreover, I suggest to extract the necessary data to present the novel discoveries.

Response: The abstract was rewritten focusing in the topic discussed in the manuscript.

Reviewer:  There are serious mistakes  and flaws in this manuscript e.g. a second messenger seems to be interchanged with receptors.The manuscript has to be rewritten completely and the English to be improved. There are lots of typos and grammatical errors. The sentences are too long which makes it difficult to follow up. The titles of the legends are not clear and the legends themselves are too long.

Response: The manuscript was carefully revised and several errors in verbal agreement were corrected. 

Reviewer:  Parts of the conclusions have to be replaced into the introduction. Conclusively, I cannot recommend this manuscript for publication.

 Response:  The conclusion was modified, and some sentences were moved to introduction.

Reviewer: Minor Comments: 

Response: All minor comments were carefully corrected in the manuscript.

Reviewer 3 Report

This is a very extensive review of a growing research field, for which a comprehensive report appears timely.  The manuscript is well written and addresses comprehensively some complex ideas associated with these cyclic dinucleotides. 

It is imperative however that the authors spend a little time highlighting the chemical structure of these cyclic dinucleotide for which very simple difference in the nucleobase results in important specific biology. The chemistry should be covered early in the text even if their biosynthesis is covered later on in the manuscript. 

The summary table which highlights function should be edited so that only key points are noted rather than sentences. 

There are some redundancies in some part of the manuscript which could be avoided. 

Author Response

Dear Reviewer 3,

Thanks a lot for your careful reading of the manuscript and for the constructive suggestions.

Following are the answers for your questions or suggestions.

Reviewer:  It is imperative however that the authors spend a little time highlighting the chemical structure of these cyclic dinucleotide for which very simple difference in the nucleobase results in important specific biology. The chemistry should be covered early in the text even if their biosynthesis is covered later on in the manuscript. 

Response: The introduction has been rewritten with more CDN chemistry information. Line 41: "Recently, this research area has been under expansion, with the discoveries of new intracellular signaling cyclic dinucleotides (CDNs) in bacteria. In 2008, it was demonstrated that bacteria can produce not only c-di-GMP, but also c-di-AMP, cyclic-bis(3′ → 5′)-dimeric AMP, by an enzyme known as DisA that possess a DAC domain [3]. In 2012, a novel cyclic dinucleotide has been found to be a second bacterial messenger, cGAMP, cyclic guanosine (3' → 5') monophosphate - adenosine (3' → 5') monophosphate, synthesized by proteins containing SMODS domain such as the DncV protein [4, 5]. At the moment, c-di-GMP, c-di-AMP and c-GAMP have been described as the main bacterial second messengers. Nevertheless, different classes of cyclic oligonucleotides, such as c-UAMP, c-di-UMP, c-UGM, c-CUMP and c-AAGMP, have been also found in bacteria [2, 3, 5, 6]. These molecules include not only di-purines but also hybrids of purine and pyrimidines and cyclic trinucleotides [6].

The cyclisation between two nucleotides of the most common bacterial CDNs involves the formation of a phosphodiester bond that links the C3’ of one pentose ring with the C5’ of another, resulting in a 3’ - 5’ cyclic dinucleotide (3' → 5'). Despite their chemical similarities, there are specific enzymes involved in the synthesis and degradation of different CDNs. Furthermore, bacteria have different classes of CDN receptors that are specific to only one type of CDN. However, how the receptors differentiate one CDN from another is still unclear. Given the specificity of the receptor, since this is the molecule responsible for directly or indirectly regulating different bacterial phenotypes, changes in a single base of the CDN can lead to quite divergent biological responses, as described below."

Reviewer: The summary table which highlights function should be edited so that only key points are noted rather than sentences. 

Response: We tried to summarize Table 1 as much as we could, but unfortunately, we cannot reformulate the table 1.

Reviewer: There are some redundancies in some part of the manuscript which could be avoided.

Response: The manuscript has been carefully revised, and redundancies were avoided. 

Reviewer 4 Report

The manuscript is an extensive revision of the present knowledge on the production and metabolism of cyclic dinucleotides especially centered in structural studies of enzymes involved in bacterial metabolism. It is an excellent review on the role and metabolism of cyclic dinucleotides which have a very important role in intracellular signaling. Cyclic dinucleotides act as second messengers regulating a large number of bacterial functions such as bacterial motility, colony and biofilm formation and others. The review is well written and contains a large number of detailed information. The subject is of large interest and the manuscript has an excellent quality providing extensive information of enzymatic systems that may be of interest for a large number of structural biologists. There is only a few minor comments.

Some Figures are difficult to read. For example the formulas that represent chemical reactions at the bottom of Figure 1 are not readable. This part should be more clear and bigger. Figure 7 also is difficult to read, especially the alignment of sequences in Figure 7H and some parts. Increasing the size could improve the readability Figure 8 A also contain chemical formulas that are not clear. In general I recommend to the authors to dedicate some time to improve the readability of the information present in the figures.

Minor grammatical errors:

Legend of figure 1 at the end (line 103) it says” FAPy: 2-amino-5-Figure6. 5’-phosphate” and it should be “FAPy: 2-amino-5-formylamino-6-ribosylamino-4(3H)-pyrimidinone  5’- phosphate”. Figure 6 should not be there.

Page 3 , line 109, “cannot be describe” should be ““cannot be described”

Page 4, receptor function column, “Nevertheless, it is controversy” should be “Nevertheless, it is controversial

Page 5, receptor function column, “protein complex that synthesis” should be “protein complex that synthesize”

Page 5, receptor function column, “The bind of “ should be “The binding of”

Page 10, receptor function column, “FleQ is a transcription regulator and contains three domains …. a interaction domains”, should be FleQ is a transcription regulator and a contain three domains ….a interaction domain”.

Page 11, receptor function column, “STING proteins interacts” should be “STING proteins interact”

Page 13, receptor function column, it says “Kd range of 0.1 to 8 M”. Molar is very high I believe it should be milimolar, or micromolar or nanomolar. Please check

Page 14, receptor function column, it says “Kd range of 4.8 M”. Molar is very high I believe it should be milimolar, or micromolar or nanomolar. Please check

Page 15, line 167, “GGDEF domains synthesizes” should be “GGDEF domains synthesize”

Page 21, line 340, legend figure 5, it says “ATP are found bond” should be “ATP are found bound”

Page 23, line 624-625, it says “The magnesium ion is coordinate by” and it should be “The magnesium ion is coordinated by”

Page 23 line 655. It seem that “Active site of DAC Domains” should be underlined as the previous subheading “DAC structure…(line 629).

Page 25, line 687. Check the heading style : Cyclic di-nucleotide receptors as it is in Capital Letters but the other headings are in italics. The same with the next two headings in pages 28 and 31.

Page 27, line 761, it says “It is controverse” and it should be “It is controversial”

Page 27, line 765, it says “Bioinformatics studies” and it should be “Bioinformatic studies”

Page 33, line 1005, it says “but can be also used as a stimulator of the innate immune system”.

I believe it should be in plural as the previous phrase is in plural, it should be “but can be also used as stimulators of the innate immune system”.

References. Please check references 19 (no page numbers), 116, 131, 155, 158,177 (no journal name, no year, no page number, no volume number)

Author Response

Dear Reviewer 4,

Thanks a lot for your careful reading of the manuscript and for the constructive suggestions. Following are the answers for your questions or suggestions.

Reviewer: Some Figures are difficult to read. For example the formulas that represent chemical reactions at the bottom of Figure 1 are not readable. This part should be more clear and bigger. Figure 7 also is difficult to read, especially the alignment of sequences in Figure 7H and some parts. Increasing the size could improve the readability Figure 8 A also contain chemical formulas that are not clear. In general I recommend to the authors to dedicate some time to improve the readability of the information present in the figures.

Response: I agree completely with the reviewer. The images were replaced with high resolution. 

Reviewer: Minor grammatical errors:

Response: All minor grammatical errors as well as the references that were incomplete have been corrected.